# Optimization-by-design of hepatotropic lipid nanoparticles targeting the sodium-taurocholate cotransporting polypeptide

Dominik Witzigmann[1,2]*, Philipp Uhl[3], Sandro Sieber[1], Christina Kaufman[3,4], Tomaz Einfalt[1], Katrin Schöneweis[4], Philip Grossen[1], Jonas Buck[1], Yi Ni[4], Susanne H Schenk[1], Janine Hussner[5], Henriette E Meyer zu Schwabedissen[5], Gabriela Québatte[1], Walter Mier[3], Stephan Urban[4], Jörg Huwyler[1]*

[1]Division of Pharmaceutical Technology, Department of Pharmaceutical Sciences, University of Basel, Basel, Switzerland; [2]Department of Biochemistry and Molecular Biology, University of British Columbia, Vancouver, Canada; [3]Department of Nuclear Medicine, University Hospital Heidelberg, Heidelberg, Germany; [4]Department of Infectious Diseases, Molecular Virology, University Hospital Heidelberg, INF, Heidelberg, Germany; [5]Division of Biopharmacy, Department of Pharmaceutical Sciences, University of Basel, Basel, Switzerland

**Abstract** Active targeting and specific drug delivery to parenchymal liver cells is a promising strategy to treat various liver disorders. Here, we modified synthetic lipid-based nanoparticles with targeting peptides derived from the hepatitis B virus large envelope protein (HBVpreS) to specifically target the sodium-taurocholate cotransporting polypeptide (NTCP; *SLC10A1*) on the sinusoidal membrane of hepatocytes. Physicochemical properties of targeted nanoparticles were optimized and NTCP-specific, ligand-dependent binding and internalization was confirmed in vitro. The pharmacokinetics and targeting capacity of selected lead formulations was investigated in vivo using the emerging zebrafish screening model. Liposomal nanoparticles modified with 0.25 mol% of a short myristoylated HBV derived peptide, that is Myr-HBVpreS2-31, showed an optimal balance between systemic circulation, avoidance of blood clearance, and targeting capacity. Pronounced liver enrichment, active NTCP-mediated targeting of hepatocytes and efficient cellular internalization were confirmed in mice by [111]In gamma scintigraphy and fluorescence microscopy demonstrating the potential use of our hepatotropic, ligand-modified nanoparticles.
DOI: https://doi.org/10.7554/eLife.42276.001

*For correspondence:
dominik.witzigmann@unibas.ch (DW);
joerg.huwyler@unibas.ch (JöH)

**Competing interests:** The authors declare that no competing interests exist.

## Introduction

The design of hepatotropic drug carriers is of great interest for the treatment of various liver disorders (*Williams et al., 2014*; *Poelstra et al., 2012*; *Reddy and Couvreur, 2011*). In particular if cell-type specific delivery of macromolecular therapeutic agents, selective targeting of parenchymal liver cells and internalization is needed. Previously, hepatocyte targeted nanoparticles have been developed exploiting endogenous and exogenous targeting ligand-based mechanisms using glycan, protein or antibody modifications of the nanoparticle surface (*Akinc et al., 2010*; *Akinc et al., 2009*; *Barrett et al., 2014*; *Detampel et al., 2014*; *Witzigmann et al., 2016a*). Most established systems for liver-specific drug delivery rely on targeting the hepatic asialoglycoprotein (ASGPR) or low density lipoprotein (LDLR) receptors. However, studies investigating alternative targeting strategies based on other hepatocyte-specific receptors are limited. In this respect, a promising alternative might be offered by the hepatitis B virus (HBV), which shows a pronounced efficacy to infect the human liver due to its strong affinity to hepatocytes. Less than 10 virus particles have been shown to

be sufficient to efficiently target hepatocytes of chimpanzees resulting in a pathogenic HBV infection (*Asabe et al., 2009*). The reason for its extraordinary liver tropism is a highly specific amino acid sequence in the large HBV envelope protein (*i.e.* HBVpreS1 domain), which is essential for target receptor recognition (*Meier et al., 2013*; *Schieck et al., 2013*). For decades, the specific target of HBV on the sinusoidal membrane of hepatocytes was unknown until in 2012 the interaction with the human sodium-taurocholate cotransporting polypeptide (NTCP/SLC10A1) was identified (*Yan et al., 2012*). Subsequently, Urban and colleagues performed a fine mapping of the HBVpreS sequence to identify the amino acids responsible for efficient binding (*Schulze et al., 2010*; *Ni et al., 2014*; *Schieck et al., 2013*). As a result, the first HBV/HDV entry inhibitor, a myristoylated peptide named Myrcludex B, was developed and successfully introduced in clinics (currently phase II clinical trials) (*Blank et al., 2016*; *Urban et al., 2014*). Myrcludex B binds with high affinity and specificity to human NTCP on the sinusoidal membrane of hepatocytes thereby blocking binding of virus particles to their target cells.

Based on these findings, the question arises whether Myrcludex B might serve as a targeting ligand to design a hepatotropic, NTCP-specific nanoparticle. In recent years, several groups have therefore attempted to develop targeting strategies based on HBV envelope proteins, for example recombinant HBV envelope protein particles (bio-nanocapsules) or HBV preS1-derived functionalized liposomes (*Liu et al., 2016*; *Somiya et al., 2016*; *Somiya et al., 2015*; *Zhang et al., 2015*; *Zhang et al., 2014*). However, the nanoparticulate drug delivery systems developed had physicochemical properties (e.g. size, colloidal stability, and immunogenic potential), which were sub-optimal for efficient in vivo targeting of hepatocytes. Especially the size of the nano-formulations presented a limitation. Most developed formulations had sizes above the average diameter of hepatic fenestrations in healthy humans (i.e. 100 nm) (*Wisse et al., 2008*) thereby limiting the passage through liver fenestrations and consequently the access to the space of Disse and the sinusoidal membrane of hepatocytes. Notably, the liver fenestrae diameter of rodents show high species and strain differences ranging from around 100 nm to 160 nm, possibly explaining positive liver targeting of published formulations (*Braet and Wisse, 2002*; *Steffan et al., 1987*; *Wisse et al., 2008*). In addition, a nanoparticle size above 100 nm triggers phagocytosis by cells of the reticuloendothelial system (*i.e.* hepatic Kupffer cells and spleen macrophages) resulting in rapid blood clearance (*Kettiger et al., 2013*). Both factors significantly decrease the likelihood of reaching the parenchymal liver tissue and increase the risk for potential off-target effects in untargeted tissues.

Surface properties are another important characteristic of nanoparticles. The surface charge (i.e. ζ potential) should be slightly negative (*Xiao et al., 2011*) to prevent sequestration of particles in the lung (*i.e.* due to a positive charge) (*Ishiwata et al., 2000*) or rapid clearance by cells expressing scavenger receptors (*i.e.* due to an excessive negative charge) (*Rothkopf et al., 2005*). According to the classical Derjaguin-Landau-Verwey-Overbeek (DLVO) theory of colloids, a neutral charge has to be avoided to prevent particle agglomeration. In addition to surface charge, steric stabilization by PEGylation mediates long circulating properties and prevents opsonization (*Karmali and Simberg, 2011*; *Milla et al., 2012*).

It was the aim of the present study to design and optimize a nanoparticle based on liposomes combined with derivatives of Myrcludex B to efficiently target hepatocytes while minimizing interactions with off-target cell types. Optimization of physicochemical properties of the nanoparticles included size and charge optimization and steric shielding by PEGylation. Derivatives of Myrcludex B were selected based on target binding, cellular uptake and their impact on the colloidal stability of nanoparticles. For the lipid membrane composition, we used a FDA and EMA approved multi-component lipid formulation based on Doxil (*i.e.* liposomal formulation of doxorubicin) (*Barenholz, 2012*). To design an optimal targeted system, several Myrcludex B derivatives with variations in the peptide sequence or fatty acid modification were covalently linked to the distal end of PEG-lipids. NTCP-specific and ligand-dependent uptake was confirmed in vitro using human liver-derived cell lines. Recently, Shan et al. reported huge discrepancies between in vitro systems and rodent experiments during the development of targeted nanomedicines (*Shan et al., 2015*). Therefore, we used the zebrafish as a complementary in vivo screening model based on our previous work (*Sieber et al., 2019b*; *Campbell et al., 2018*; *Einfalt et al., 2018*; *Sieber et al., 2017*). We assessed the effect of nanoparticles' ligand type and ligand density on their pharmacokinetics. To this end, human-derived cell lines lacking or expressing the human NTCP (hNTCP, *SLC10A1*) were xenotransplanted into zebrafish embryos prior to systemic administration of

nanoparticles. Finally, tissue distribution of dual-labeled nanoparticles was qualitatively (fluorescence-based) and quantitatively (radionuclide-based) investigated in vivo in mice to demonstrate the targeting potential of our hepatotropic nanoparticle platform in higher vertebrates.

## Results and discussion

### Design and characterization of a hepatotropic nanoparticle for NTCP-specific targeting

The aim of our study was the design of a hepatotropic, targeting ligand-modified nanoparticle. To this end, the surface of liposomal nanoparticles was modified using targeting peptides or lipopeptides derived from the preS1 domain of the HBV large envelope protein (*Figure 1A*). Based on a previous screening of 26 HBVpreS peptide variants, we selected Myrcludex B, the first HBV entry inhibitor (*Blank et al., 2016*; *Bogomolov et al., 2016*), and five additional Myrcludex B-derived peptides to evaluate the influence of amino acid sequence variations or acyl chain modifications on targeting efficiency and thereby optimize our hepatotropic nanoparticle. All Myrcludex B derived (lipo) peptides were synthesized in high yields and purity by standard solid phase peptide synthesis using Fmoc-chemistry (*Schieck et al., 2013*; *Schieck et al., 2010*; *Müller et al., 2013*). Lipopeptides were N-terminally modified with the fatty acids myristic acid (saturated C14) or capric acid (saturated C10), since our previous studies have shown that fatty acid modification is key for mediating interactions with target cells. C-termini of synthesized targeting (lipo)peptides were modified with cysteine residues to allow conjugation to the distal end of PEGylated phospholipids (DSPE-PEG2000-Maleimide) integrated into sterically stabilized liposomes. Coupling was achieved by a chemically reactive maleimide, giving rise to a metabolically stable thioether bond suitable for applications in living organisms (*Figure 1A*). Successful conjugation of Myrcludex B to lipid-based nanoparticles was demonstrated by fluorescence correlation spectroscopy using Myrcludex B-Atto488. The autocorrelation curve of nanoparticle conjugated peptides showed a significant shift to longer diffusion times as compared to the free peptide, with average diffusion times of $\tau_d$ = 1639 μs and $\tau_d$ = 192 μs, respectively (*Figure 1—figure supplement 1*).

Liposome membrane partition coefficients of mono fatty acid modifications are orders of magnitudes lower as compared to di-lipid anchors. (*Sauer et al., 2006*) Therefore, the distearoyl anchor of DSPE results in a stable incorporation of the PEGylated phospholipid-targeting ligand conjugate in the lipid bilayer of liposomes (membrane partition coefficient $>10^3$ mM$^{-1}$), whereas the PEG linker offers a flexibility to the distally tethered lipopeptides to extend away from the liposome surface. In addition, a thermodynamically favorable backward bending insertion of the acyl chain into the liposomal membrane is possible. A slight change in transition temperature evaluated by pressure perturbation calorimetry and differential scanning calorimetry confirmed this hypothesis (data not shown). The formulation yield of modified nanoparticles after purification was dependent on the conjugated Myrcludex B derived (lipo)peptide with preS2−48 > Myr-preS2−31 > Myr-preS2-48A $\geq$ Myr-preS2−48 > Cap-preS2-48.

Light scattering and electron microscopy verified that all nanoparticles, that Myrcludex B-derived peptide conjugated liposomes (modified without or with C14 acyl moiety) had a spherical morphology with a small size around 90 nm, narrow size distribution (*that is* PDI <0.2), and a slightly negative zeta potential (*Figure 1B*, *Table 1*). Only a small increase in the hydrodynamic size of about 2 nm was observed after conjugation of Myrcludex B-derived peptides (*Table 1*). The zeta potential of nanoparticles remained negative due to a negative net charge of Myrcludex B-derived lipopeptides at physiological pH. Thus, the physicochemical properties of nanoparticles were not significantly influenced by the surface modification with HBVpreS derived lipopeptides containing C14 acyl chains. Exceptions were nanoparticles modified with Cap-preS2-48, which had an average diameter of 134.28 nm and a PDI of 0.24 (*Figure 1B*, *Table 1*).

It is tempting to speculate, that the C10 acyl chain of Cap-preS2-48 interfered with liposome membrane stability. As compared to longer acyl chains the backward bending insertion of C10 acyl chains into intra-liposomal membranes is less stable, thus promoting faster dissociation and possible interactions with neighboring liposomes due to re-association with inter-liposomal membranes. This was also indicated by formation of aggregates resulting in shorter storage stability (data not shown). Previously published studies reporting a rapid partitioning of shorter lipid anchors from liposomal

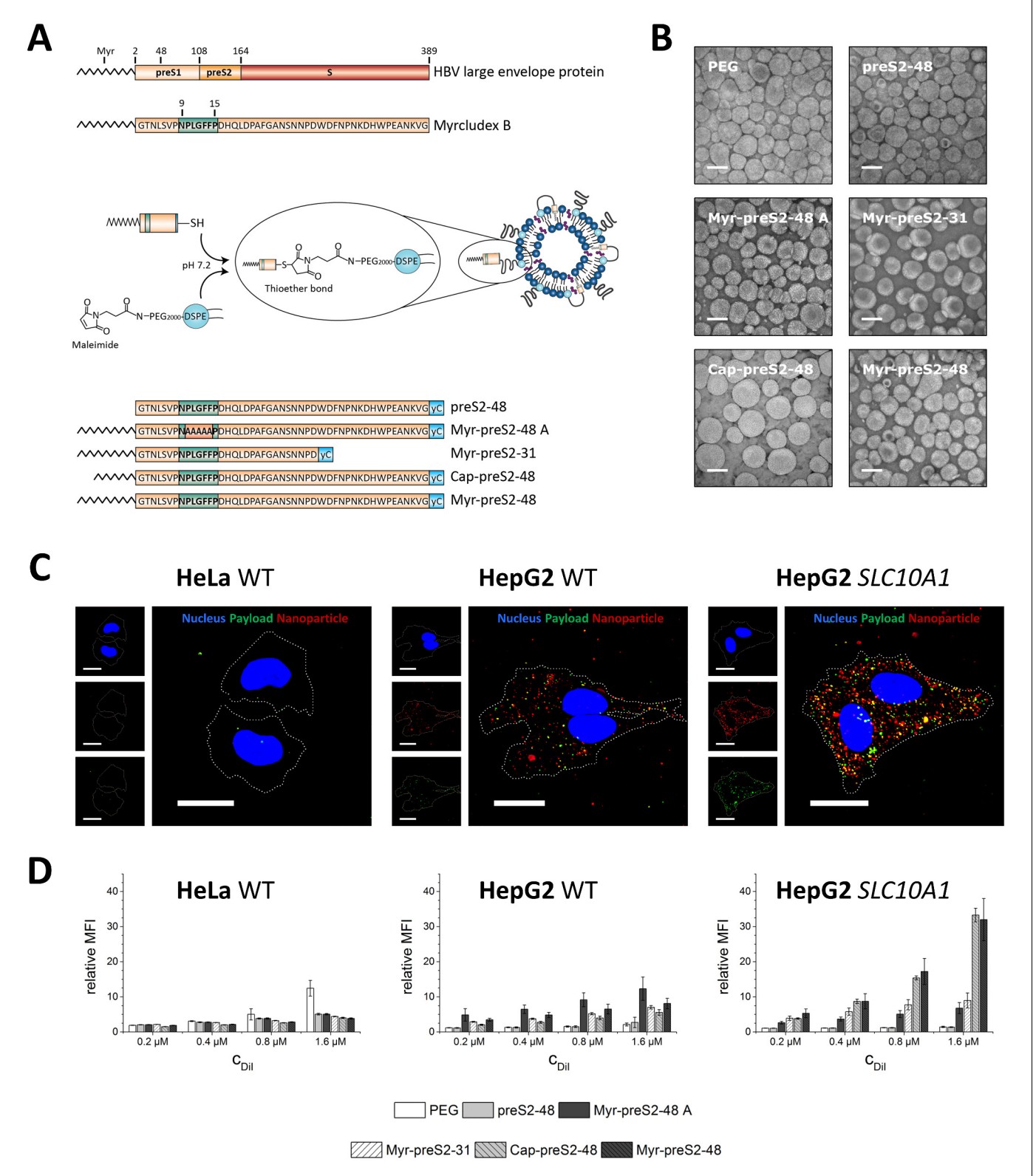

**Figure 1.** Hepatotropic nanoparticles based on liposomes modified with Myrcludex B-derived peptides for NTCP-specific targeting. (**A**) Schematic representation of peptides derived from Hepatitis B virus (HBV) large envelope protein including the first entry inhibitor, Myrcludex B. Different peptides were conjugated via thiol function to the distal end of PEG chains integrated in the nanoparticle structure using maleimide chemistry. The most important amino acid sequence (9-15) for NTCP-specific binding is highlighted in green color in each lipopeptide. (**B**) Representative transmission

*Figure 1 continued on next page*

*Figure 1 continued*

electron microscopy images of different Myrcludex B-derived lipopeptide conjugated nanoparticles. Scale bar = 100 nm. (C) Uptake of Myrcludex B-modified nanoparticles into human cells with variable NTCP expression levels. Nanoparticles have a dual fluorescent label, that is lipophilic membrane label (DiI, red) and hydrophilic payload incorporation (carboxyfluorescein, green). Representative confocal laser scanning microscopy maximum intensity projections for Myr-preS2-48 modified nanoparticles after 30 min are shown. Dotted lines indicate cell membranes. Blue signal: Hoechst stain of cell nuclei. Scale bar = 10 μm. (D) Flow cytometry analysis of uptake rate into non-hepatic HeLa cells, liver derived HepG2 cells and *SLC10A1* overexpressing HepG2 cells. Increasing concentrations of nanoparticles ($C_{DiI}$) modified with different Myrcludex B-derived peptides were evaluated. Relative mean fluorescence intensities (MFI) of DiI signals normalized to untreated cells are given. All values are shown as mean ± SD of biological replicates (n ≥ 3 independent experiments). Numerical data for all graphs are shown in *Figure 1—source data 1*.
DOI: https://doi.org/10.7554/eLife.42276.002

The following source data and figure supplements are available for figure 1:

**Source data 1.** Characterization of hepatotropic nanoparticles.
DOI: https://doi.org/10.7554/eLife.42276.012

**Figure supplement 1.** Characterization of hepatotropic nanoparticles based on liposomes modified with Myrcludex B (Myr-preS2-48) using fluorescence correlation spectroscopy.
DOI: https://doi.org/10.7554/eLife.42276.003

**Figure supplement 2.** Assessment of cytocompatibility of nanoparticles modified with different Myrcludex B derived peptides using non-hepatic HeLa cells (HeLa WT), liver-derived wildtype HepG2 cells (HepG2 WT) and HepG2 cells overexpressing the human NTCP (HepG2 *SLC10A1*) by MTT assay.
DOI: https://doi.org/10.7554/eLife.42276.004

**Figure supplement 3.** Concentration dependent fluorescence self-quenching of 5 (6)-carboxyfluorescein.
DOI: https://doi.org/10.7554/eLife.42276.005

**Figure supplement 4.** Flow cytometry analysis of nanoparticle uptake rate into non-hepatic HeLa cells, liver-derived HepG2 cells and *SLC10A1* overexpressing HepG2 cells.
DOI: https://doi.org/10.7554/eLife.42276.006

**Figure supplement 5.** Uptake of Myrcludex B-modified nanoparticles into HuH7 liver-derived cells deficient (HuH7 WT) or overexpressing *SLC10A1* (HuH7 *SLC10A1*).
DOI: https://doi.org/10.7554/eLife.42276.007

**Figure supplement 6.** Time-dependent internalization of Myr-preS2-31 modified nanoparticles into *SLC10A1* overexpressing HepG2 cells.
DOI: https://doi.org/10.7554/eLife.42276.008

**Figure supplement 7.** Time-dependent internalization and toxicity of propidium iodide loaded nanoparticles into *SLC10A1* overexpressing HepG2 cells.
DOI: https://doi.org/10.7554/eLife.42276.009

**Figure supplement 8.** Time-dependent internalization and toxicity of doxorubicin loaded nanoparticles into *SLC10A1* overexpressing HepG2 cells.
DOI: https://doi.org/10.7554/eLife.42276.010

**Figure supplement 9.** Activity of DNA loaded lipid nanoparticles (LNP).
DOI: https://doi.org/10.7554/eLife.42276.011

**Table 1.** Physicochemical characteristics of nanoparticles with different surface modifications. Hydrodynamic size [nm], polydispersity index (PDI), and zeta potential [mV] were analyzed using dynamic and electrophoretic light scattering. All values are shown as mean ± SD of n ≥ 3 independent experiments. Numerical data for all nanoparticles are shown in *Table 1—source data 1.*.

| Surface modification | Size [nm] ± SD | PDI ± SD | Zeta potential [mV] ± SD |
|---|---|---|---|
| PEG | 88.53 ± 5.89 | 0.05 ± 0.01 | −5.93 ± 0.63 |
| preS2-48 | 90.74 ± 5.83 | 0.06 ± 0.02 | −3.34 ± 1.38 |
| Myr-preS2-48 A | 90.77 ± 4.98 | 0.06 ± 0.04 | −13.35 ± 3.08 |
| Myr-preS2-31 | 89.10 ± 4.38 | 0.10 ± 0.02 | −9.82 ± 0.87 |
| Cap-preS2-48 | 134.28 ± 36.23 | 0.24 ± 0.04 | −8.39 ± 1.13 |
| Myr-preS2-48 | 92.21 ± 6.78 | 0.12 ± 0.08 | −10.70 ± 4.25 |

DOI: https://doi.org/10.7554/eLife.42276.013
The following source data is available for Table 1:
**Source data 1.** Physicochemical characterization of nanoparticles.
DOI: https://doi.org/10.7554/eLife.42276.014

vesicles support our observation of unfavorable liposome interactions for Cap-preS2-48 (*Sauer et al., 2006*; *Webb et al., 1998*).

## Cellular uptake and viability

Next, we investigated the biocompatibility and targeting capacity of our ligand-modified nanoparticles in a panel of three different cell lines in vitro, that is non-hepatic HeLa cells devoid of *SLC10A1* (negative control), liver-derived wild type HepG2 cells (HepG2 WT, hepatocyte control cell line with no detectable *SLC10A1* expression based on PCR) and HepG2 cells overexpressing the human NTCP (HepG2 *SLC10A1*). We used lentiviral transduced cells overexpressing *SLC10A1* as a positive control to confirm the specificity of our system since human liver derived cell lines such as HepG2 and HuH7 down-regulated NTCP during oncogenic transformation (i.e. NTCP expression levels are significantly decreased in hepatocellular carcinoma) (*Lempp et al., 2016*). In all cell lines, nanoparticles showed a high cytocompatibility up to the highest tested lipid concentration of 8 mM, which is far beyond liposome blood concentrations achievable in a clinical setting (*Figure 1—figure supplement 2* demonstrating no decrease of cell viability using the MTT assay) (*Barpe et al., 2010*).

In vitro uptake studies revealed that Myrcludex B-modified nanoparticles were rapidly internalized within 30 min into liver-derived HepG2 cell lines whereas no binding or cellular uptake was observed in non-hepatic HeLa cells (*Figure 1C*, representative confocal laser scanning microscopy images for Myr-preS2-48 modified nanoparticles). Both the liposomal nanoparticle (DiI signal) and the encapsulated payload (carboxyfluorescein (CF) signal), were detected intracellularly. Notably, CF was encapsulated into our nanoparticles at a fluorescence self-quenching concentration (i.e. 60 mM, *Figure 1—figure supplement 3*). Thus, CF fluorescence increases significantly after overcoming the Förster critical transfer distances, that is release of CF from nanoparticles into surrounding environment (*Chen and Knutson, 1988*). Specific uptake of nanoparticles with NTCP-binding component preS-peptide was enhanced with increasing *SLC10A1* expression levels (*Figure 1C,D* and *Figure 1—figure supplement 4*) demonstrating a high target specificity (*i.e.* HeLa WT <HepG2 *SLC10A1*). Surprisingly, the highest DiI signal (and not CF signal) in HeLa cells was observed with PEGylated nanoparticles. It is tempting to speculate that nanoparticle modification with Myrcludex B-derived lipopeptides decreases the interaction with negatively charged cell membranes of *SLC10A1*-deficient cells (*e.g.* HeLa) due to a negative net charge of lipopeptides at physiological pH and thus increased electrostatic repulsion. Uptake studies with a different liver-derived cell line (HuH7) comparing wild type and *SLC10A1* overexpressing cells confirmed the *SLC10A1* specific interaction. Overexpression of *SLC10A1* again resulted in a strong enrichment of cellular uptake ruling out an involvement of cell-line specific artifacts (*Figure 1—figure supplement 5*).

In order to demonstrate the potential application of Myr-preS2-31 modified nanoparticles as drug delivery system, we successfully incorporated small molecular payloads as well as larger compounds into nanoparticles payloads (i.e. propidium iodide, doxorubicin, FITC-labeled peptide, DNA vector) to enhance their internalization into *NTCP* expressing cells (*Figure 1—figure supplements 6*, *7*, *8* and *9*). Indeed, time-dependent uptake studies confirmed the rapid binding and internalization process of Myr-preS2-31 modified nanoparticles (*Figure 1—figure supplements 6*, *7* and *8*). Of note, propidium iodide is a cell membrane impermeable drug. Thus, NTCP-targeted nanoparticles enabled internalization into cells and successful release into cytosol indicated by nuclear counterstain (*Figure 1—figure supplement 7*). To investigate the potential application of *NTCP*-targeted lipid nanoparticles as gene delivery systems, we encapsulated a DNA vector coding for GFP into lipid nanoparticles based on a clinically approved lipid composition and modified with Myr-preS2-31. High content screening analysis demonstrated that modification of nanoparticles with Myr-preS2-31 significantly increases the transfection of *NTCP* expressing cells (*Figure 1—figure supplement 9*). These experiments highlight future applications of the developed carriers and serve as a starting point for future extended in vivo studies in different species and disease models.

## Competition of NTCP-specific cellular binding and uptake of targeting ligand-modified nanoparticles

To confirm specificity of NTCP interactions with Myrcludex B derived ligands, we used pre-incubations with free Myrcludex B-Atto565 to competitively inhibit nanoparticle binding and cellular uptake (*Figure 2A*). Fluorescently labeled Myrcludex B can be considered to be a suitable blocking agent

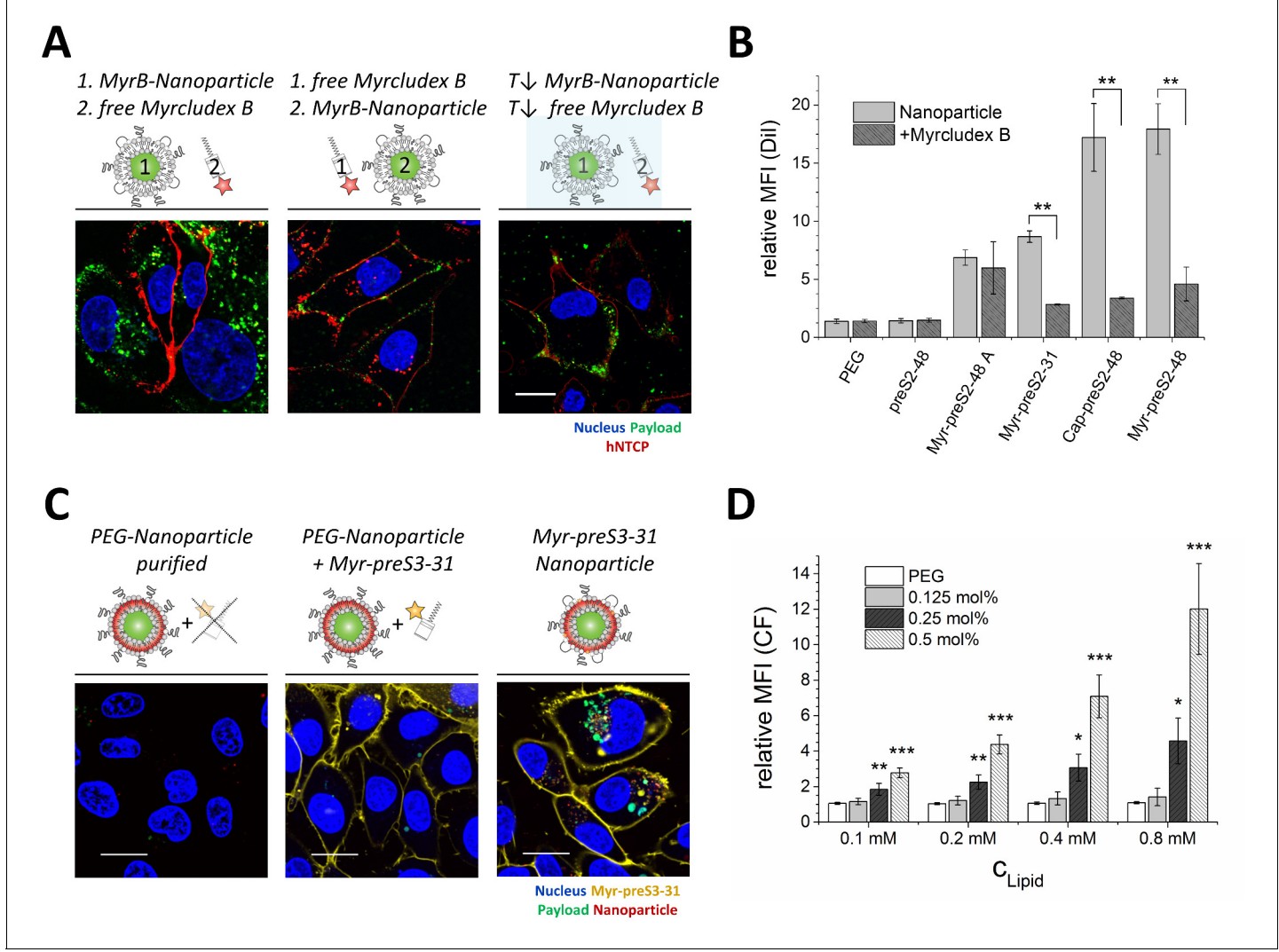

**Figure 2.** NTCP-specific and ligand-dependent uptake of Myrcludex B-derived lipopeptide conjugated nanoparticles into liver-derived cells in vitro. (A) Competitive inhibition study of Myrcludex B (MyrB)-conjugated nanoparticle uptake (carboxyfluorescein payload, green signal) into HepG2 *SLC10A1* already after 30 min. Free Atto-565 labeled Myrcludex B (red signal) was added after (left panel) or before (middle panel) nanoparticle. Uptake studies at lower temperature (T↓, 4°C, right panel) were performed to demonstrate energy-dependent process of nanoparticle internalization. Representative confocal laser scanning microscopy images are shown. Blue signal: Hoechst stain of cell nuclei. Scale bar = 10 μm. (B) Quantification of nanoparticle uptake in absence or presence of free Myrcludex B dependent on different Myrcludex B derived peptide modification. All values are shown as mean ± SD of biological replicates (n = 3 independent experiments). **p<0.01. (C) Uptake study of nanoparticles (membrane dye, DiI, red signal) loaded with carboxyfluorescein (green signal) into HepG2 *SLC10A1* without, mixed or covalently modified with Atto-633 conjugated Myr-preS2-31 (yellow signal). Myr-preS2-31-K-Atto633 is covalently linked to surface via stable thioether bond (right panel). Representative confocal laser scanning microscopy images are shown. Blue signal: Hoechst stain of cell nuclei. Scale bar = 10 μm. (D) Concentration ($C_{Lipid}$) dependent uptake of nanoparticles modified with different amounts of Myr-preS2-31 analyzed by flow cytometry and based on CF signal. All values are shown as mean ± SD of biological replicates (n = 4 independent experiments). *p<0.05, **p<0.01, ***p<0.001. Numerical data for all graphs are shown in *Figure 2—source data 1*.
DOI: https://doi.org/10.7554/eLife.42276.015

The following source data and figure supplements are available for figure 2:

**Source data 1.** Influence of ligand on cellular interactions.
DOI: https://doi.org/10.7554/eLife.42276.021

**Figure supplement 1.** NTCP-dependent uptake mechanism of nanoparticles.
DOI: https://doi.org/10.7554/eLife.42276.016

**Figure supplement 2.** Uptake of Myrcludex B-modified nanoparticles into HeLa cells transfected with empty vector (pEF6 Ctrl), *Slc10a1* or *SLC10A1*.
DOI: https://doi.org/10.7554/eLife.42276.017

**Figure supplement 3.** Influence of glycosaminoglycans (GAGs) on Myrcludex B binding.

*Figure 2 continued on next page*

*Figure 2 continued*

DOI: https://doi.org/10.7554/eLife.42276.018

**Figure supplement 4.** Uptake of nanoparticles into HepG2 WT cells in absence or presence of heparan sulfate.

DOI: https://doi.org/10.7554/eLife.42276.019

**Figure supplement 5.** Ligand density dependent uptake of Myrcludex B-modified nanoparticles loaded with carboxyfluorescein (payload, green).

DOI: https://doi.org/10.7554/eLife.42276.020

since its binding to NTCP expressing HepG2 *SLC10A1* cells results in a significant shift in fluorescence signal as compared to control cells (data not shown). Uptake inhibition of nanoparticles modified with Myr-preS2-31, Cap-preS2-48, and Myr-preS2-48 by free Myrcludex B-fluorescein was confirmed by flow cytometry (*Figure 2B*). In contrast, the uptake of Myr-preS2-48A modified nanoparticles was not significantly inhibited by free Myrcludex B, due to the non-specific amino acid sequence (see difference in essential amino acid sequence highlighted in *Figure 1A*). By incubation of cells in presence of NaN$_3$ or at low temperature (i.e. 4°C), we confirmed that the uptake of NTCP targeted nanoparticles is an energy-dependent process (*Figure 2A*). These results demonstrate that hepatotropism of nanoparticles is mediated by NTCP and that the cellular uptake of the carrier is an active and energy-dependent process.

## Selection of the optimal hepatotropic Myrcludex B-derived lipopeptide

After evaluating the formulation yield, physicochemical characteristics (i.e. storage/colloidal stability, hydrodynamic diameter, size distribution, zeta potential) and the targeting capacity of our NTCP-specific nanoparticles in vitro, we identified Myr-preS2-48 and Myr-preS2-31 as lead structures and used these for further investigations. This choice was based on the following observations:

First, only nanoparticles modified with lipopeptides but not peptides without conjugated fatty acid (e.g. preS2-48) can bind to NTCP. This set of experiments confirmed that the acyl modification of peptides on nanoparticles' surface is a crucial prerequisite for hepatocyte binding as reported recently for free peptides (*Meier et al., 2013*; *Schieck et al., 2013*). Only acyl modified peptides increased nanoparticle binding and internalization. Second, liposomes decorated with peptides conjugated to capric acid had a reduced colloidal stability. Their storage stability was limited due to particle aggregation. Furthermore, their size of around 134 nm exceeds the diameter of liver sinusoid fenestrations presumably limiting their access to the space of Disse. Third, Myr-preS2-48A modified nanoparticles were excluded due to poor NTCP specificity as demonstrated by the lack of binding competition by free Myrcludex B. In addition, the uptake of these nanoparticles was independent of NTCP expression levels and even higher in HepG2 wild type cells (*Figure 1D*).

## Mechanistic studies on NTCP mediated cellular binding and internalization

In order to demonstrate the importance of covalent peptide attachment, we used a triple fluorescence labeling strategy (*Figure 2C*). The targeting ligand Myrcludex B was labeled with Atto633, the liposomal phospholipid bilayer was labeled with DiI, and the aqueous cargo payload of nanoparticles consisted of CF. Myr-preS2-31-K-Atto633 was labeled at an additionally introduced lysine at position 2, in order to still allow conjugation to the nanoparticle surface by the terminal cysteine. Recently, we have shown that additional N-terminal amino acids do not interfere with specific liver enrichment (for comparison Myr-preS−11-48) (*Schieck et al., 2013*).

First, CF-loaded, DiI-labeled nanoparticles were incubated with Myr-preS2-31-K-Atto633 and purified using size exclusion chromatography to remove free targeting ligand. Cell experiments confirmed successful removal of free Myr-preS2-31-K-Atto633 (no signal on cell membrane) and as expected no uptake of PEGylated nanoparticles. As a control, we added a mixture of free Myr-preS2-31-K-Atto633 and PEGylated nanoparticles to HepG2 *SLC10A1* cells without prior purification. Notably, a strong fluorescence signal on the cell membrane was observed due to specific binding of free Myr-preS2-31-K-Atto633 to *SLC10A1* indicating specific targeting despite an additional N-terminal amino acid. Free Myr-preS2-31-K-Atto633 did not interact with PEGylated nanoparticles

and thus did not trigger nanoparticle entry into HepG2 *SLC10A1* cells. Finally, we covalently linked the Myr-preS2-31-K-Atto633 to the nanoparticle surface by Michael addition of the distal cysteine residue to maleimide-functionalized PEGylated phospholipids integrated in the nanoparticle structure. A strong cellular binding and uptake of Myr-preS2-31-K-Atto633 modified nanoparticles was observed already within 1 h.

Interestingly, nanoparticles including their payload entered the target cell whereas the targeting ligand remained on the cell surface. Since Myrcludex B has a remarkably high affinity to the NTCP ($K_D$ of 67 nM) (*Meier et al., 2013*), it is tempting to speculate that the targeting ligand is retained by NTCP on the cell surface while the dissociated liposome payload is internalized and further processed by a yet unknown mechanism. Of note, intracellular CF signals were considerably higher when compared to DiI signals. This might also indicate liposome dissociation and perhaps loss of DiI during the internalization process. Uptake experiments using pharmacological pathway inhibitors suggested a partially clathrin-dependent and caveolin-independent mechanism, which differs from the process of phagocytosis and micropinocytosis (*Figure 2—figure supplement 1*). Intriguingly, additional factors besides NTCP binding seem to contribute to this process. Non-hepatic HeLa cells transduced with mouse NTCP (mNtcp; *Slc10a1*) or *SLC10A1* can bind Myrcludex B-modified nanoparticles. However, binding is reduced as compared to binding in hepatic cell lines and no uptake is observed (*Figure 2—figure supplement 2*). Thus, additional hepatic cell dependent factors seem to play a role for efficient binding and internalization. Indeed, Verrier et al. reported recently that glypican five expression is an important co-factor for HBV entry (*Verrier et al., 2016*). Notably, uptake experiments using psgA745 cells (CHO xylosyltransferase mutants) overexpressing NTCP showed that binding of Myrcludex B alone is not influenced by glycosaminoglycans (*Figure 2—figure supplement 3*). In sharp contrast, binding of nanoparticles could be partially inhibited in HepG2 WT cells using heparan sulfate suggesting an involvement of glycosaminoglycans in the binding and subsequent internalization process of nanoparticles for hepatic cells (*Figure 2—figure supplement 4*). Therefore, it will be an important step for the design of next-generation carrier systems to elucidate such co-factors in detail and adapt the nano-sized delivery system accordingly.

To demonstrate concentration-dependent nanoparticle uptake of Myr-preS2-31 and investigate the effect of ligand density, qualitative and quantitative fluorescence techniques were used (*Figure 2D*). Therefore, we performed in vitro experiments using nanoparticles with variable amounts of coupled Myr-preS2-31 (0 mol% - 0.5 mol% initial maleimide functionalities on nanoparticle surface). With increasing targeting ligand concentration, a significant increase in cellular uptake was observed (*Figure 2D*, *Figure 2—figure supplement 5*). Notably, we identified a threshold value of at least 0.25 mol% for efficient cell binding by qualitative confocal imaging as well as quantitative flow cytometry experiments (*Figure 2D*, *Figure 2—figure supplement 5*). Below this value, no uptake was observed, whereas above 0.25 mol% cellular binding was improved. Stoichiometric estimations assuming a bilayer thickness of 5 nm, a phosphatidylcholine headgroup area of 0.71 nm$^2$ and an equal distribution of DSPE-PEG in the outer and inner nanoparticle membrane result in $157 \pm 16$, $79 \pm 8$, or $39 \pm 4$ maleimide moieties per liposome capable for lipopeptide conjugation corresponding to 0.5 mol%, 0.25 mol%, or 0.125 mol%, respectively (*Maurer et al., 2001*). Thus, a minimum of 80 functional maleimide moieties per nanoparticle is necessary for efficient cellular targeting after Myr-preS2-31 conjugation.

## In vivo systemic circulation in the zebrafish vertebrate model

Since in vitro experimental models are not able to mimic the physiological complexity of nano-bio interactions at an organ level, we screened in the next step the effect of ligand density on pharmacokinetics of nanoparticles in vivo for Myr-preS2-48 and Myr-preS2-31 (*Figure 3*). Recently, we have reported that the zebrafish is a valuable pre-clinical tool to assess the systemic circulation and blood clearance of nanoparticulate drug delivery systems in vivo (*Sieber et al., 2017*; *Campbell et al., 2018*; *Sieber et al., 2019b*; *Park, 2017*; *Yin et al., 2018*; *Sieber et al., 2019a*).

Thus, we injected DiI labeled nanoparticles modified with different amounts of targeting ligand (0.125 mol% - 1.0 mol%) into the duct of Cuvier of transgenic kdrl:EGFPs843 zebrafish embryos which express GFP in the vasculature endothelial cells. Already 1 h post injection, a clear qualitative difference in circulation characteristics of tested nanoparticles was detected. With increasing ligand density on the nanoparticle surface, the systemic circulation of nanoparticles decreased for both peptides (*Figure 3*) indicating that ligand modification of nanoparticles interferes with the shielding

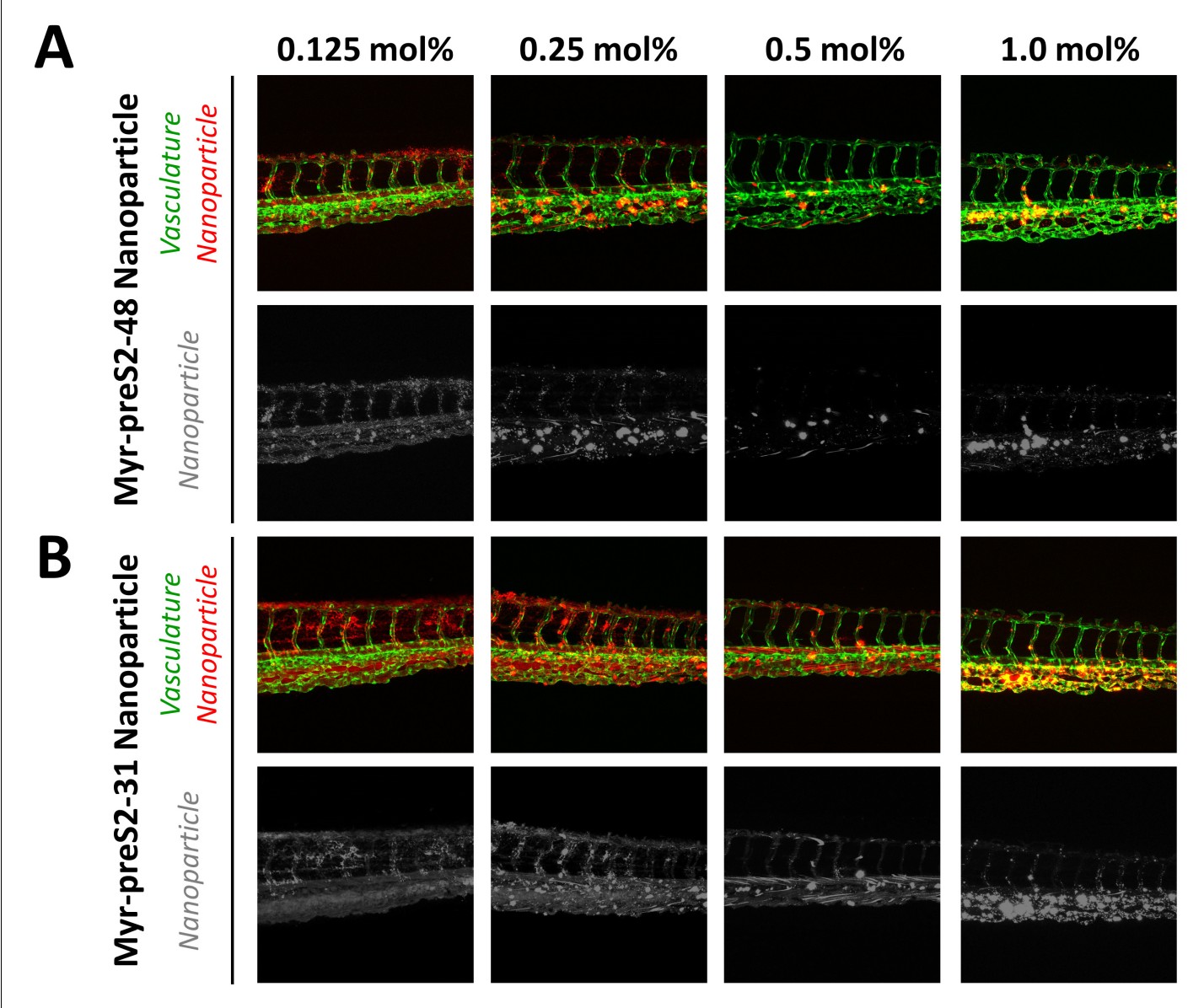

**Figure 3.** Systemic circulation of Myrcludex B-derived lipopeptide conjugated nanoparticles in vivo in the zebrafish model. Nanoparticles were modified with different amounts of (**A**) Myr-preS2-48 or (**B**) short Myr-preS2-31 and injected into transgenic zebrafish embryos expressing green fluorescent protein in their vasculature endothelial cells (green signal). Membrane of nanoparticles was fluorescently labeled using DiI (red signal). Representative confocal laser scanning microscopy images of tail region 1 hpost injection.
DOI: https://doi.org/10.7554/eLife.42276.022

properties of PEG. Increased blood clearance was thereby paralleled by accumulation in the posterior caudal vein region. The observed binding pattern did not match a *stabilin-2* scavenger receptor dependent nanoparticle clearance, which would be indicative for interactions with mammalian liver sinusoidal endothelial cells (LSECs) (*Campbell et al., 2018*). More likely a sequestration by macrophages is responsible for this clearance mechanism corresponding to an accumulation in the spleen of rodents (*Sieber et al., 2019b*).

Interestingly, nanoparticles modified with the shorter targeting peptide, that is Myr-preS2-31, showed increased systemic circulation (*Figure 3B*) as compared to Myr-preS2-48 modified nanoparticles (*Figure 3A*) at similar ligand densities. Thus, Myr-preS2-31 modified nanoparticles were selected for further investigations. However, nanoparticles modified with more than 0.5 mol% Myr-preS2-31 were as well excluded from further evaluation due to their poor systemic circulation and high clearance rate.

## In vivo targeting ability in the zebrafish vertebrate model

In a next step, we investigated the targeting capacity of Myr-preS2-31 modified nanoparticles to human cells in vivo in the zebrafish model (*Figure 4*). In recent years, several groups have used xenografted zebrafish for various investigations including the assessment of nanoparticles (*Sieber et al.,*

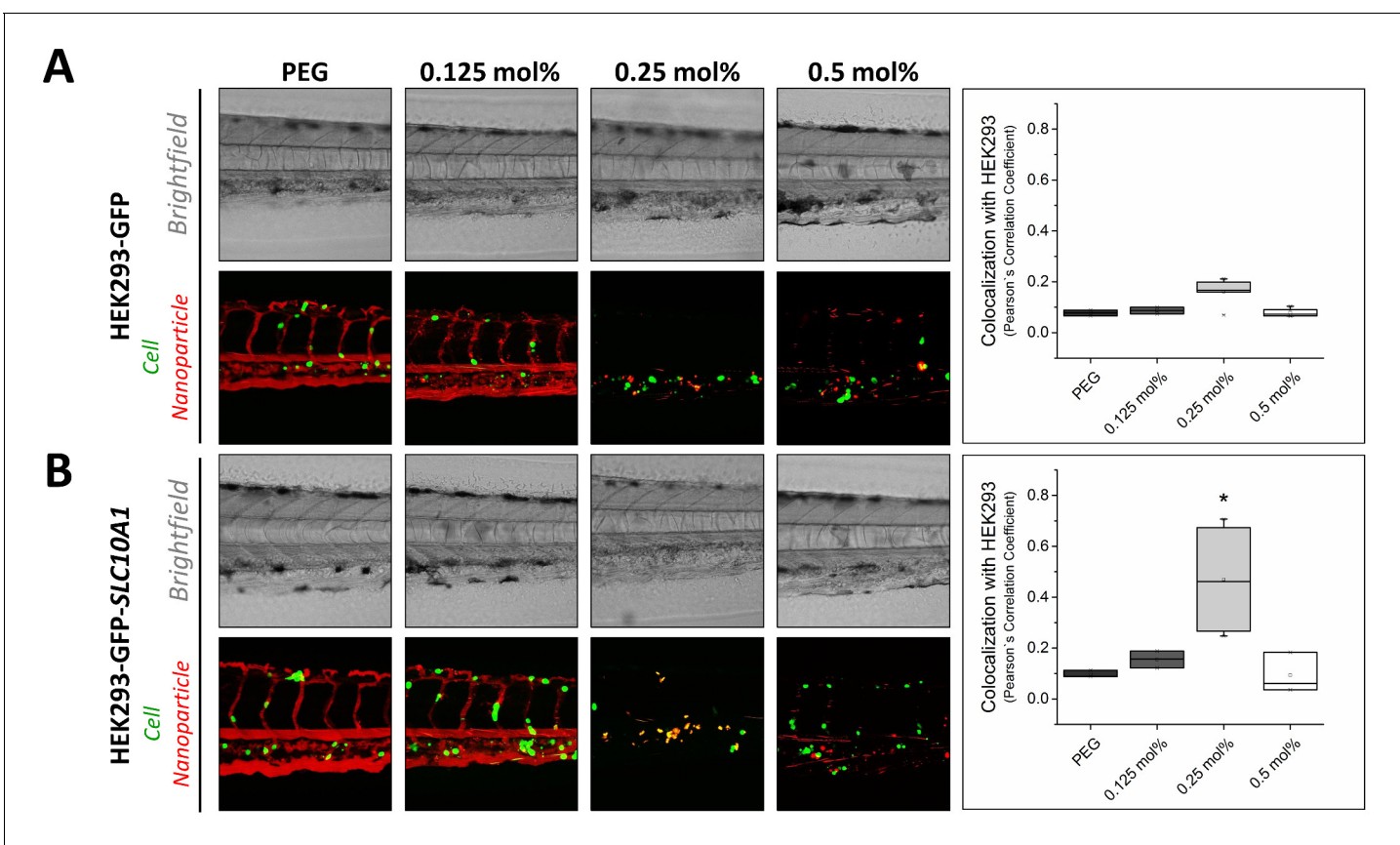

**Figure 4.** Targeting ability of Myr-preS2-31 conjugated nanoparticles in vivo in xenotransplanted zebrafish embryos. Nanoparticles were modified with different amounts of Myr-preS2-31 and injected into wild type zebrafish embryos xenotransplanted with human, GFP expressing HEK293 cells (green signal), (A) deficient or (B) expressing *SLC10A1*. Membrane of nanoparticles was fluorescently labeled using DiI (red signal). Yellow signals demonstrate colocalization (*i.e.* binding and internalization) of nanoparticles with HEK293-GFP cells. Representative brightfield and fluorescence images of tail region 1 h post injection are shown. Quantitative analysis of nanoparticle binding to HEK293-GFP cells is represented by Pearson´s Correlation Coefficient (PCC). All values are shown as box plots of biological replicates (n ≥ 2 independent experiments). *$p<0.05$. Numerical data for all graphs are shown in *Figure 4—source data 1*.

DOI: https://doi.org/10.7554/eLife.42276.023

The following source data and figure supplement are available for figure 4:

**Source data 1.** Targeting xenotransplanted cells in the zebrafish model.
DOI: https://doi.org/10.7554/eLife.42276.025
**Figure supplement 1.** Targeting ability of free Myrcludex B in vivo in xenotransplanted zebrafish embryos.
DOI: https://doi.org/10.7554/eLife.42276.024

*2019a*; *Evensen et al., 2016*; *Wertman et al., 2016*; *Brown et al., 2017*; *He et al., 2012*; *Lin et al., 2017*; *Veinotte et al., 2014*; *Wagner et al., 2010*). Despite anatomical differences with mammals, zebrafish xenotransplantation models are an emerging preclinical tool offering several practical advantages as compared to mouse xenografting models including prolific reproduction, facilitated xenotransplantation (no immune rejection due to limited adaptive immune response), and optical transparency enabling high throughput screening. For our study, we used HEK293 cells stably expressing GFP for further genetic modification and establishment of xenotransplants (*Witzigmann et al., 2015b*). HEK293-GFP cells were transiently transfected with *SLC10A1* to express the targeting factor for our hepatotropic nanoparticles. Wild type HEK293-GFP without *SLC10A1* served as control. Both cell lines were injected into ABC/TU wild type zebrafish embryos to create human xenotranplants. The different nanoparticles were injected as soon as trans-genic human cells stopped circulating and remained in the caudal vasculature tail region (*i.e.* after approximately 1 h). Interestingly, a clear difference in targeting capacity dependent on *SLC10A1* expression and ligand density was revealed. Whereas there was no significant difference in targeting capacity at different ligand densities for *SLC10A1*-deficient HEK293-GFP cells (*Figure 4A*), a significant increase in binding to HEK293-GFP cells was observed if *SLC10A1* was overexpressed as the nanoparticles could bind specifically and be readily internalized (*Figure 4B*). Most importantly, this was only valid for nanoparticles modified with 0.25 mol% Myr-preS2-31 (*Figure 4*, quantitative analysis). This illustrates that ligand density highly influences the balance between systemic circulation, systemic clearance rate and targeting efficiency of our liposome-based nanoparticles.

Nanoparticles modified with ligand densities below 0.25 mol% show a favorable systemic circulation but have an insufficient targeting ability. This also confirms our observations in vitro, where nanoparticles with a ligand density below 0.25 mol% did not significantly bind to HepG2 *SLC10A1* cells. In sharp contrast, nanoparticles modified with higher Myr-preS2-31 targeting ligand densities (*i.e.* 0.5 mol%) have increased targeting ability in vitro. However, decreased systemic circulation and a high clearance rate under in vivo conditions counteract the advantage of higher ligand densities. Nanoparticles modified with 0.25 mol% Myr-preS2-31 have the highest targeting efficiency due to an ideal balance between target affinity and long circulation time in vivo. It should be noted that nanoparticles are internalized by target cells (*Figure 4B*) whereas free Myrcludex B apparently binds with high affinity to target cells but is not internalized (*Figure 4—figure supplement 1*). This phenomenon was recently observed by our team in rodents (data not shown) and was also reported from clinical trials in humans.

## In vivo liver targeting of Myr-preS2-31 conjugated nanoparticles in mice

To elucidate the influence of ligand density on hepatotropism of our nanoparticles in vivo in mammals, we evaluated the pharmacokinetic properties of Myr-preS2-31 conjugated nanoparticles in mice. For this set of experiments, we used a dual labeling approach. The radioactive nuclide indium-111 ($^{111}$In) was used for whole-body imaging and biodistribution studies whereas fluorescence labeling with DiI was used to evaluate intra-organ nanoparticle distribution. Importantly, we incorporated DTPA-conjugated DSPE into the lipid bilayer to chelate $^{111}$In on the surface of nanoparticles. This radiolabeling strategy has distinct advantages as compared to other labeling techniques or loading of $^{111}$In-oxine into nanoparticles (*van der Geest et al., 2015*). First, this radiolabeling method is robust, fast (within 45 min) and efficient with labeling efficiencies above 90%. Notably, free $^{111}$In was easily removed from nanoparticle formulations prior to injection using size exclusion chromatography (NAP-5 columns). Second, DTPA-DSPE enables retention of $^{111}$In in serum for at least 48 h at 37°C (>98% label retention) demonstrating the high stability necessary for in vivo studies of nanoparticulate drug delivery systems (*van der Geest et al., 2015*). Third, free $^{111}$In is rapidly eliminated via kidneys and excreted in the urine as shown previously (*Harrington et al., 2000*; *Shih et al., 2017*). This offers an easy assessment to differentiate between non-bound and nanoparticle bound $^{111}$In.

Four different lipid-based nanoparticles were prepared and injected intravenously into the tail vein of mice, that is PEGylated liposomes (negative control) and nanoparticles modified with 0.125 mol%, 0.25 mol% and 0.5 mol% Myr-preS2-31. Plasma and organs were harvested to perform a quantitative biodistribution analysis ex vivo 1 h post injection (*Figure 5A*). PEGylated nanoparticles showed the typical biodistribution of sterically stabilized nanoparticles with a strong signal in the blood (*Figure 5A*). Myr-preS2-31 conjugated liposomes demonstrated different biodistribution patterns depending on ligand density (*Figure 5A*). Modification of nanoparticles with 0.125 mol% Myr-

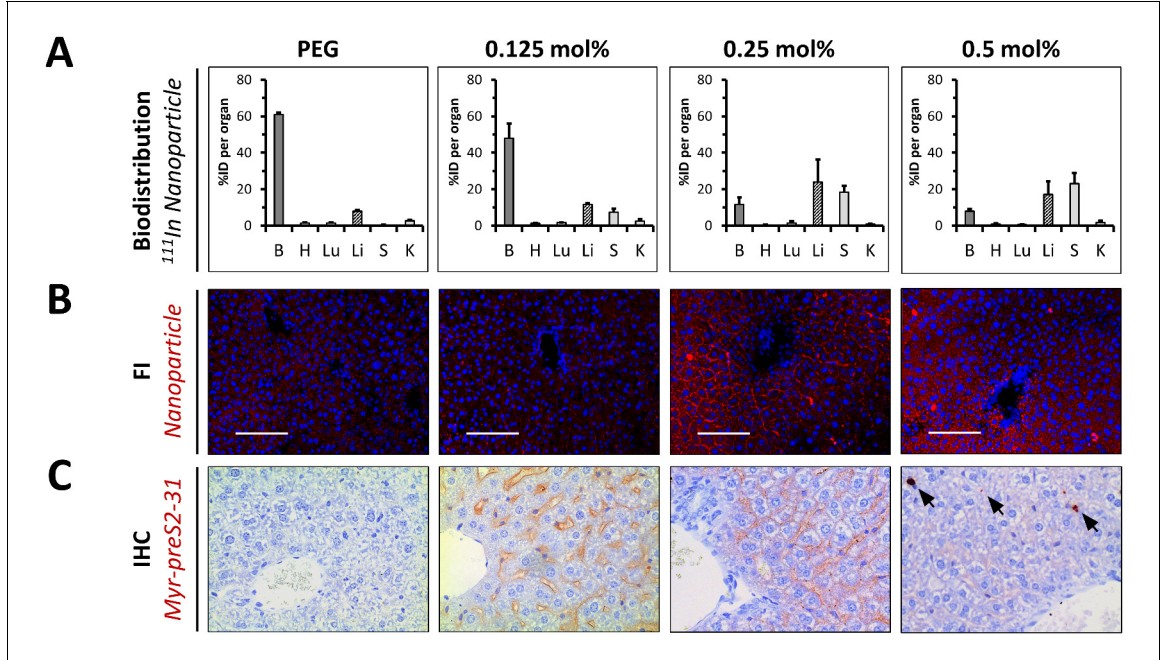

**Figure 5.** In vivo biodistribution and liver targeting of Myr-preS2-31 conjugated nanoparticles in mice. Nanoparticles were modified with different amounts of Myr-preS2-31 and labeled with radioactive [111]In and fluorescent membrane dye (DiI, red signal). (**A**) Quantitative biodistribution studies were performed 1 h post injection. Radioactivity of each organ was determined with a γ-counter and the percentage of injected dose (%ID) per organ was calculated. B = blood, H = heart, Lu = lung, Li = liver, S = spleen, K = kidney. All values are shown as mean ± SD of biological replicates (n = 4 independent experiments). Numerical data for all graphs are shown in *Figure 5—source data 1*. (**B**) Fluorescence imaging (FI) of nanoparticles (DiI, red signal) in liver cryo-sections. Blue signal: Hoechst stain of cell nuclei. Scale bar = 100 μm. (**C**) Immunohistochemistry (IHC) of Myr-preS2-31 (red signal) in the liver sections 1 h after intravenous injection. Mice liver sections were stained with anti-Myr-preS2-31 antibody (MA18/7). Blue signals represent cell nuclei. Arrows indicate distinct localized accumulations.
DOI: https://doi.org/10.7554/eLife.42276.026

The following source data and figure supplements are available for figure 5:

**Source data 1.** Biodistribution in mice.
DOI: https://doi.org/10.7554/eLife.42276.036
**Figure supplement 1.** In vivo biodistribution and liver targeting of Myr-preS2-31 conjugated nanoparticles in mice.
DOI: https://doi.org/10.7554/eLife.42276.027
**Figure supplement 2.** In vivo biodistribution and ex vivo organ distribution of PEGylated nanoparticles in mice.
DOI: https://doi.org/10.7554/eLife.42276.028
**Figure supplement 3.** In vivo biodistribution and ex vivo organ distribution of nanoparticles with elevated Myr-preS2-31 modification.
DOI: https://doi.org/10.7554/eLife.42276.029
**Figure supplement 4.** Organ biodistribution of different nanoparticles in rats.
DOI: https://doi.org/10.7554/eLife.42276.030
**Figure supplement 5.** Organ ratios of ex vivo biodistribution analysis in mice.
DOI: https://doi.org/10.7554/eLife.42276.031
**Figure supplement 6.** Intra-organ distribution of Myr-preS2-31-modified nanoparticles in spleen and kidney in vivo in mice dependent on ligand density.
DOI: https://doi.org/10.7554/eLife.42276.032
**Figure supplement 7.** Liver targeting of Myr-preS2-31 conjugated nanoparticles in mice.
DOI: https://doi.org/10.7554/eLife.42276.033
**Figure supplement 8.** Specific binding of Myr-preS2-31 conjugated nanoparticles to sinusoidal membrane of hepatocytes.
DOI: https://doi.org/10.7554/eLife.42276.034
**Figure supplement 9.** Biodistribution of nanoparticles modified with 0.5 mol% Myr-preS2-31 in mice.
DOI: https://doi.org/10.7554/eLife.42276.035

preS2-31 did not alter the systemic circulation significantly (*i.e.* high blood pool signal). Only a minor increase in liver accumulation was observed as compared to ligand-lacking PEGylated nanoparticles (*Figure 5A*).

Interestingly, nanoparticles modified with 0.25 mol% Myr-preS2-31 significantly enriched binding to the liver (*Figure 5A*). Further increase in ligand density (0.5 mol%) resulted in an increase in spleen accumulation, that is enhanced clearance by cells of the reticuloendothelial system (*Figure 5A*). Of note, none of the nanoparticle formulation resulted in an elimination via kidneys demonstrating the high stability and retention of the DTPA-DSPE chelated $^{111}$In.

Planar gamma scintigraphy imaging of injected mice and harvested organs at various time points (15 min and 60 min) confirmed these observations (*Figure 5—figure supplements 1*, *2* and *3*). PEGylated nanoparticles demonstrated the typical systemic circulation with a strong signal in highly perfused organs, for example heart (*Figure 5—figure supplements 1* and *2*). Modification of nanoparticles with 0.25 mol% Myr-preS2-31 significantly increased the liver accumulation. Interestingly, further increase in Myr-preS2-31 modification ($\geq$0.25 mol%) resulted in dominant location of radioactivity in an elongated structure in the far-left part of the abdomen under the liver, which was identified as the spleen (*Figure 5—figure supplement 1*). In order to confirm this observation, we injected mice with excessive Myr-preS2-31 modified nanoparticles (>0.5 mol%) and performed a planar imaging (*Figure 5—figure supplement 3*). Indeed, elevated Myr-preS2-31 modification resulted in an exclusive spleen accumulation, that is enhanced clearance by cells of the reticuloendothelial system. In order to exclude species specific effects, we also performed a planar gamma scintigraphy imaging of harvested organs from injected rats (*Figure 5—figure supplement 4*). Again, elevated Myr-preS2-31 modification has negative impacts on liver accumulation.

In order to highlight the ligand-density dependent hepatotropism, we calculated ratios between the blood pool and important organs, that is liver (*i.e.* target organ), spleen (*i.e.* clearance organ), and kidney (*i.e.* control organ since nanoparticle bound $^{111}$In should not show renal excretion) (*Figure 5—figure supplement 5*). Indeed, Myr-preS2-31 modification $\geq$0.25 mol% resulted in increased liver/spleen-to-blood ratios. Strikingly, nanoparticles modified with 0.25 mol% Myr-preS2-31 demonstrated a significant increase in the liver-to-kidney ratio confirming our conclusions from the zebrafish model, that 0.25 mol% Myr-preS2-31 is the optimal ligand density (*Figure 4B*).

The biodistribution studies were combined with fluorescence imaging of nanoparticle distribution (*Figure 5B*, *Figure 5—figure supplement 6*) and immunohistochemistry of Myr-preS2-31 distribution (*Figure 5C*, *Figure 5—figure supplements 6–9*) in liver, spleen, and kidney (*i.e.* nanoparticles and Myr-preS2-31 should not show renal excretion). PEGylated nanoparticles showed a weak fluorescent signal in the liver (*Figure 5B*). Importantly, these signals were not associated with the sinusoidal membrane of hepatocytes but arose from the high hepatic blood supply (*Figure 5—figure supplement 8*). No signals were observed in spleen and kidney (*Figure 5—figure supplement 6*). Modification of nanoparticles with 0.125 mol% Myr-preS2-31 did not result in significantly increased liver levels. A marginal binding of nanoparticles to hepatocyte membrane was visually observed (*Figure 5B,C*). This supports our hypothesis that a threshold level of targeting ligand density present on the nanoparticle surface is necessary for successful targeting. Importantly, strong signals for nanoparticles modified with 0.25 mol% Myr-preS2-31 were observed on the basolateral membrane of parenchymal liver cells (*Figure 5B,C*) demonstrating the strong hepatotropism of our nanoparticles. Further increasing the ligand density (*i.e.* 0.5 mol%) was detrimental and resulted in a diffuse hepatic staining pattern. Nanoparticles and their payload were detected as punctuated signals in the whole liver and did not show a specific membrane staining (*Figure 5B*, *Figure 5—figure supplement 9*). Myr-preS2-31 was detected in distinct localized areas only (*Figure 5C*). We conclude that nanoparticles modified with excessive Myr-preS2-31 densities (0.5 mol%) are rapidly cleared by liver resident macrophages, *i.e.* Kupffer cells. Subsequent re-distribution phenomena result in an unspecific nanoparticle signal in the whole liver.

## Competition of NTCP-specific uptake into mouse hepatocytes in vivo

Since nanoparticles modified with 0.25 mol% Myr-preS2-31 allowed highly efficient liver targeting, we next investigated the NTCP specificity and the internalization process (*Figure 6A*). Therefore, we injected either labeled nanoparticles alone or together with free unlabeled Myrcludex B into mice (*Figure 6B*). Co-injection of Myrcludex B resulted in a clear decrease in liver enrichment by competitive inhibition of NTCP-binding as demonstrated by a change of signal.

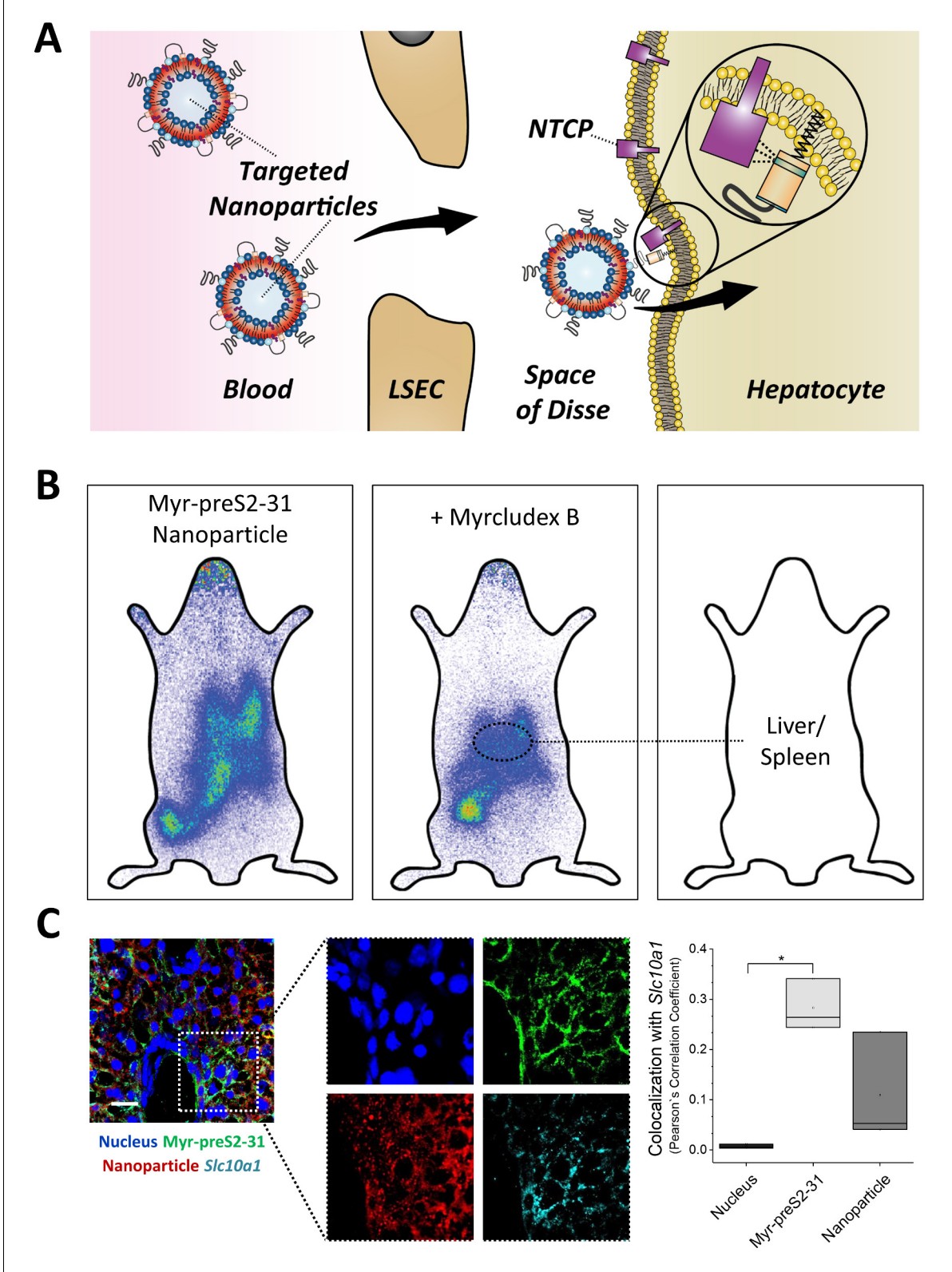

**Figure 6.** NTCP-specific uptake of Myr-preS2-31 conjugated nanoparticles into hepatocytes in vivo in mice. (**A**) Schematic representation of NTCP-targeted nanoparticle binding to hepatocytes. Circulating nanoparticles pass the fenestrae of liver sinusoidal endothelial cells (LSEC) and subsequently bind to the NTCP in the basolateral membrane of hepatocytes facing the space of Disse. Prior to Myr-preS2-31 mediated NTCP binding, the myristoyl chain is inserted into the lipid bilayer. In close proximity to hepatocytes, acyl chain switches into cellular membrane due to high affinity of essential

*Figure 6 continued on next page*

*Figure 6 continued*

peptide sequence to NTCP binding site, thereby consolidating target transporter binding. (**B**) Nanoparticles were modified with 0.25 mol% of Myr-preS2-31 and labeled with radioactive $^{111}$In and fluorescent membrane dye (DiI, red signal). Planar imaging of mice 1 h post-injection. Competitive inhibition of liver binding by co-injection of free Myrcludex B clearly demonstrates NTCP-specific binding. Positions of liver and spleen are indicated by small circles. (**C**) Immunofluorescence staining of nanoparticles (red signal), targeting ligand (Myr-preS2-31, green signal, antibody staining), and *Slc10a1* (cyan signal, antibody staining) in liver cryosections. Nuclei staining (blue signal) served as control for complete internalization; no overlap with *Slc10a1*. Scale bar = 20 µm. Colocalization analysis with *Slc10a1* is represented by Pearson´s Correlation Coefficient (PCC). All values are shown as box plots of biological replicates (n = 3 independent experiments). *p<0.05. Numerical data for all graphs are shown in *Figure 6—source data 1*.

DOI: https://doi.org/10.7554/eLife.42276.037

The following source data is available for figure 6:

**Source data 1.** Internalization into hepatocytes.

DOI: https://doi.org/10.7554/eLife.42276.038

To reveal the localization of nanoparticles, we performed a confocal microscopy analysis of liver cryo-sections (*Figure 6C*). We stained liver cryo-sections using antibodies against *Slc10a1* and Myr-preS2-31 (MA 18/7). Interestingly, nanoparticles that were internalized into parenchymal liver cells did not co-localize with *Slc10a1* fluorescent signals. Myr-preS2-31 still colocalized with *Slc10a1* suggesting that the targeting ligand was separated from the nanoparticle during cellular internalization as already observed in in vitro experiments. This phenomenon was confirmed by a colocalization analysis (*Figure 6C*). The observed cellular uptake is a surprising finding since HBV possesses pronounced host species specificity with regard to binding and infectivity. HBV binds to murine hepatocytes but cannot infect mice due to the lack of host cell dependency factors (*Lempp et al., 2016*). Therefore, chimeric mice transplanted with primary human hepatocytes have been developed to study anti-HBV drugs (*Petersen et al., 2008*; *Lütgehetmann et al., 2012*). In humans or chimpanzees only, HBV specifically binds to hepatocytes and subsequently infects the host.

Importantly, our NTCP-targeted nanoparticles apparently lack this species specificity. In contrast to HBV, our hepatotropic nanoparticles specifically bind to mouse hepatocytes in a *Slc10a1*-dependent manner and are subsequently internalized. The exact molecular interactions behind this internalization process will require additional studies to elucidate structural determinants important for cellular uptake and to better understand viral entry mechanisms, which are still unknown (*Glebe and Urban, 2007*).

## Conclusions

In conclusion, the combination of in vitro investigations, the zebrafish model and in vivo experiments in rodents offered a unique approach to optimize our targeting ligand modified nanoparticles. The zebrafish model demonstrated to be an excellent tool to pre-screen various nanoparticle formulations, to assess the effect of Myrcludex B modifications on their pharmacokinetics and biodistribution, and thus increase the accuracy of predictions for experiments in rodents. The developed delivery systems can increase liver uptake levels, decrease accumulation in off-target tissues and at the same time avoid clearance by the reticuloendothelial system by mimicking HBV targeting properties. Despite the fact that current liver targeting strategies such as ASGPR- or LDLR-targeted delivery systems have already demonstrated improved drug and gene delivery in various preclinical models, these systems also suffer from certain drawbacks including complex synthesis of multivalent glycans or strong evidence that a majority of endocytosed gene carriers is recycled back to the cell exterior thereby reducing activity (*Witzigmann et al., 2016a*; *Sahay et al., 2013*). Due to the availability of efficient peptide manufacturing protocols, the proposed NTCP targeted delivery platform is an alternative and promising approach which warrants further investigation. For future clinical applications, optimized Myr-preS2-31 conjugated nanoparticles entrapping small molecule drugs, nucleic acids or proteins need to be studied in appropriate animal models of disease. In particular, we see a great potential for our nanoparticle targeting strategy in the field of metabolic diseases of the liver.

# Materials and methods

**Key resources table**

| Reagent type (species) or resource | Designation | Source or reference | Identifiers | Additional information |
|---|---|---|---|---|
| Cell line (*H. sapiens*) | HepG2 WT | DOI: 10.1111/hepr.12599 | RRID:CVCL_0027 | Cell depository of the Institute of Pathology (University Hospital of Basel, Switzerland) |
| Cell line (*H. sapiens*) | HepG2 *SLC10A1* | DOI: 10.1053/j.gastro.2013.12.024 | RRID:CVCL_JY40 | Prof. Dr. Stephan Urban (University Hospital Heidelberg) |
| Cell line (*H. sapiens*) | HuH7 WT | DOI: 10.1111/hepr.12599 | RRID:CVCL_0336 | RIKEN Cell Bank (Ibaraki, Japan) |
| Cell line (*H. sapiens*) | HuH7 *SLC10A1* | DOI: 10.1053/j.gastro.2013.12.024 | | Prof. Dr. Stephan Urban (University Hospital Heidelberg) |
| Cell line (*H. sapiens*) | HeLa WT | - | RRID:CVCL_0030 | Prof. Dr. Jörg Huwyler (University of Basel) |
| Cell line (*H. sapiens*) | HEK293-GFP | DOI: 10.1021/acsami.5b01684 | | Prof. Dr. Jörg Huwyler (University of Basel) |
| Cell line (*Cricetulus griseus*) | CHO | DOI: 10.1021/bi702258z | RRID:CVCL_0214 | Prof. Dr. Joachim Seelig (University of Basel) |
| Cell line (*Cricetulus griseus*) | psgA745 | DOI: 10.1021/bi702258z | | Prof. Dr. Joachim Seelig (University of Basel) |
| Genetic reagent (*Danio rerio*) | kdrl:EGFPs843 zebrafish | DOI: 10.1016/j.jconrel.2017.08.023 | https://zfin.org/ZDB-TGCONSTRCT-070117-47 | Prof. Dr. Markus Affolter (University of Basel) |
| Antibody | anti-Myr-preS2-31 antibody (MA18/7) | - | - | Prof. Dr. Wolfram Gerlich (Justus-Liebig-Universität Gießen); monoclonal human, 1:100 dilution |
| Antibody | anti-Slc10a1 (anti-Ntcp) | - | - | Prof. Bruno Stieger (University of Zurich); polyclonal rabbit, 1:100 dilution |
| Antibody | anti-FITC antibody | Invitrogen - ThermoFisher Scientific | Catalog # 71–1900 | polyclonal rabbit, 1:100 dilution |
| Software | OriginPro 9.1 software | - | RRID:SCR_014212 | OriginLab Corporation |
| Software | FlowJo VX software | - | RRID:SCR_008520 | TreeStar |
| Software | ImageJ Fiji | - | RRID:SCR_002285 | ImageJ |

## Synthesis of Myrcludex B-derived peptides

Different peptides were synthesized by fluorenylmethoxycarbonyl/t-butyl (Fmoc/tBu) solid-phase synthesis using an Applied Biosystems 433A peptide synthesizer and modified with acyl chains as described previously (*Schieck et al., 2013*). Atto fluorescence dyes were either linked to the distal cysteine residue by maleimide chemistry or to the ε-amino group of an additionally introduced D-lysine at position two by NHS chemistry for mechanistic studies based on a triple fluorescence labeling strategy. In contrast to all other amino acids of Myrcludex B-derived lipopeptides, a D-amino acid was introduced in the latter case due to the chemical synthesis strategy used. Peptides were purified using preparative reverse-phase high performance liquid chromatography (HPLC, LaPrep P110, VWR International GmbH) with a Reprosil-Gold 120 C18 4 µm column (Dr. Maisch GmbH) and a variable gradient adapted to the peptides properties in a range of 100% $H_2O$ to 100% acetonitrile, both containing 0.1% TFA. Peptide identity was verified using an analytical Agilent 1100 HPLC system equipped with a Chromolith Performance RP-C18e column (Merck KGaA) coupled to a mass spectrometer (Exactive, Thermo Fisher Scientific).

## Preparation of hepatotropic nanoparticles

Hepatotropic nanoparticles based on liposomes were prepared using the film rehydration extrusion method as described previously (*Detampel et al., 2014*). The lipid membrane composition of nanoparticles consisted of DSPC (Lipoid AG), cholesterol (Sigma-Aldrich), DSPE-PEG2000 (Lipoid AG) at a molar ratio of 52.7:42.3:5. For the conjugation of HBV-derived peptides, DSPE-PEG2000 was replaced by DSPE-PEG2000-maleimide (Avanti Polar Lipids) at indicated molar ratios. For fluorescence labeling of lipid membrane, 1 mol% DiI (Thermo Fisher Scientific) was added to the lipid composition replacing DSPC. For radioactive labeling with [111]In, DSPC was replaced by 3 mol% DSPE-DTPA (Avanti Polar Lipids). Desired ratios of lipids were mixed; a homogenous thin film was prepared and dried using a Rotavapor A-134 (Büchi). Lipid films were rehydrated using 0.01 M PBS pH 7.2 containing 1 mM EDTA (Sigma-Aldrich) to prevent metal ion catalyzed maleimide oxidation. For passive loading and fluorescence labeling of inner core, a 60 mM 5 (6)-carboxyfluorescein (Sigma-Aldrich) solution (pH 7.4) was used for the rehydration step. At this concentration >98% of the fluorescence is self-quenched (*Figure 1—figure supplement 3*). Resulting multilamellar vesicles were subjected to five freeze-thaw cycles and extruded 11 times through a polycarbonate membrane (Avanti Polar Lipids) with a pore size of 100 nm followed by 11 times through a polycarbonate membrane with a 50 nm pore size 10°C above transition temperature (i.e. 65°C for DSPC-based formulations). For functionalization with HBV-derived peptides, nanoparticles were mixed with peptides at molar maleimide-to-cysteine ratio of 1:1 and incubated at RT overnight. To remove non-conjugated peptides and/or free hydrophilic dye, size exclusion chromatography using a Sephadex G50 column (GE Healthcare) eluted with 0.01 M PBS pH 7.4 was performed. The size exclusion chromatography column was coupled to an UV detector to analyze recovery of nanoparticles based on peak areas. Hepatotropic nanoparticles were concentrated to a final lipid concentration of 10 mM using Amicon Ultra-4 centrifugal filter units with a 100 kDa molecular weight cut-off (Millipore). DiI and cholesterol were used as marker lipids to quantify total lipid content. DiI content was quantified based on relative fluorescence signals as compared to liposome standards and in combination with Triton X-100 treatment to account for potential DiI self-quenching. Samples were excited at 561 nm and fluorescence signals were recorded using a Spectramax M2 microplate reader (Molecular Devices). The cholesterol content was determined using the Cholesterol E cholesterol assay kit from Wako following the manufacturer's protocol.

## Loading of compounds into hepatotropic nanoparticles

### FITC-peptide loading

For passive loading of FITC-Ahx-yKKEEEK into nanoparticles, a 2 mg/mL FITC-peptide solution in a mixture of PBS/DMSO/EtOH at pH 7.0 was used for the rehydration step of the homogenous lipid film. Resulting multilamellar vesicles were processed as described in the Materials and methods section.

### Propidium iodide loading

For passive loading of propidium iodide into nanoparticles, a 10 mg/mL propidium iodide solution in PBS was used for the rehydration step of the homogenous lipid film. Resulting multilamellar vesicles were processed as described in the Materials and methods section including a final purification step.

### Doxorubicin loading

For loading of doxorubicin, an active drug loading strategy based on a citrate gradient was used as previously described (*Mayer et al., 1990*). The homogenous lipid film was rehydrated using a 300 mM citrate buffer at pH 4.0 and multilamellar vesicles were processed as described in the Materials and methods section. The pH of the external buffer solution was adjusted to pH 7.0 and nanoparticles were incubated with 2 mg/mL doxorubicin at 65°C for 15 min. Free doxorubicin was removed by size exclusion chromatography.

### DNA vector loading

Lipid nanoparticles entrapping DNA were prepared as previously described with modifications (*Kulkarni et al., 2019*; *Kulkarni et al., 2018*; *Kulkarni et al., 2017*). Briefly, lipids

(ionizable lipid, cholesterol, DSPC, DSPE-PEG2000, and DSPE-PEG2000-Maleimide at a molar ratio of 50:39:10:0.75:0.25 mol%) were dissolved in ethanol at a total lipid concentration of 15 mM. The DNA vector was dissolved in 25 mM sodium acetate (pH 4.0) at an N/P ratio of 6. After T-junction mixing at a flow rate ratio of 3:1 v/v, the pH was neutralized using a 5x excess of D-PBS, the appropriate amount of Myr-preS2-31 was added, and the resulting suspension was dialyzed against D-PBS to remove residual ethanol.

## Physicochemical characterization of hepatotropic nanoparticles

### Dynamic light scattering

Size and size distribution (polydispersity index, PDI) of nanoparticles were analyzed using a Delsa Nano C Particle Analyzer (Beckman Coulter) equipped with a 658 nm laser. Samples were measured in D-PBS at RT and a measurement angle of 165˚. Data were converted using the CONTIN particle size distribution analysis (Delsa Nano V3.73/2.30, Beckman Coulter Inc).

### Electrophoretic light scattering

Zeta potential of nanoparticles was analyzed using a Delsa Nano C Particle Analyzer. Samples were measured in D-PBS at RT and a measurement angle of 15˚. Data were converted using the Smoluchowski equation (Delsa Nano V3.73/2.30).

### Transmission electron microscopy

Size and morphology of nanoparticles were analyzed using transmission electron microscopy (TEM) as described previously (*Witzigmann et al., 2015a*). In brief, samples were deposited on a 400-mesh carbon-coated copper grid, negatively stained with 2% uranylacetate, and analyzed using a CM-100 electron microscope operating at 80 kV (Philips).

### Fluorescence correlation spectroscopy

Fluorescence correlation spectroscopy (FCS) analysis of nanoparticles was performed as described previously (*Uhl et al., 2017*). In brief, Atto488, Myr-preS2-48-Atto488 and Myr-preS2-48-Atto488 conjugated nanoparticles were analyzed using an inverted confocal fluorescence laser scanning microscope (Zeiss LSM 510-META/Confocor 2) equipped with a 40 × water immersion objective lens (Zeiss C-Apochromat 40×, numerical aperture 1.2). Fluorescence intensity fluctuations were measured for three independent samples and each measurement was repeated 20–30 times. Autocorrelation functions were fitted using a two-component model and diffusion times were calculated.

## Cell culture

Cell lines were purchased from ATCC or other recognized cell depositories (Institute of Pathology, University Hospital of Basel, Switzerland and RIKEN Cell Bank, Ibaraki, Japan) who perform authentication and quality-control tests on all distribution lots of cell lines (*Witzigmann et al., 2016b*). In addition, we performed authentication tests for all cell lines based on morphology check by microscope. All human cell lines were cultured at 37˚C under 5% $CO_2$ and saturated humidity in Dulbecco's modified Eagle's culture medium high glucose (DMEM, Sigma-Aldrich) supplemented with 10% fetal calf serum (Amimed), penicillin (100 units/mL, Sigma-Aldrich), and streptomycin (100 μg/mL, Sigma-Aldrich). Stable NTCP expressing liver derived cell lines, that is HepG2 *SLC10A1* and HuH7 *SLC10A1*, were created by lentiviral transduction as published previously (*Ni et al., 2014*). For uptake experiments, different cell lines were seeded at a density of $2.5 \times 10^4$ cells/cm$^2$ and allowed to adhere for 24 h. For confocal laser scanning microscopy experiments, cells were grown on poly-D-lysine (Sigma-Aldrich) coated glass cover slips (#1.5, Menzel) or well plates (TPP).

## Transfection of cell lines

For transient expression of the transporter, plasmids encoding for mNtcp (*Slc10a1*) or hNTCP (*SLC10A1*) were generated, amplifying the coding sequence from commercially obtained mRNA (Amsbio) by PCR. The following primers were used:

   *SLC10A1*_for: 5′-ATGGAGGCCCACAACGCGTCT-3′,
   *SLC10A1*_rev 5′-CTAGGCTGTGCAAGGGGA-3′;
   *Slc10a1*_for 5′-GTGTTCACTGGGTCGGAGGATG-′3,

*Slc10a1*_rev1 5'-CAGGTCCAGAGCAAATACTCATAGGAG-'3.

Subsequently the amplicons were ligated into pEF6-V5/HIS (Invitrogen), followed by sequence verification (Microsynth). The resulting plasmids *Slc10a1*-pEF6 and *SLC10A1*-pEF6 and Lipofectamine 3000 (Sigma-Aldrich) were used for transfection of human cell lines. A standard transfection protocol was developed as follows: Plasmid DNA and P3000 reagent were diluted in Opti-MEM (Sigma-Aldrich) and rapidly mixed with Lipofectamine 3000 diluted Opti-MEM using a DNA-to-Lipofectamine 3000 w/V ratio of 3. After 5 min incubation, the transfection mix was added to adhered cells at a plasmid DNA concentration of 1 µg/mL. Control cells were either transfected with empty pEF6 vector or treated with Opti-MEM alone.

## Assessment of cytocompatibility of nanoparticles

To assess the cytocompatibility of nanoparticles modified with different Myrcludex B derived peptides a MTT cell viability assay was performed. Wild type HeLa cells, liver-derived wild type HepG2 cells and HepG2 *SLC10A1* were seeded and cultured as described above. Nanoparticles were added to cells at final concentrations of 0.25 mM – 8 mM. After 24 h, MTT reagent (Sigma-Aldrich) was added to cells for 4 h. Formazan dye crystals were solubilized for 2 h using a mixture containing 3% (v/v) sodium dodecyl sulfate (Sigma-Aldrich) and 40 mM hydrochloric acid in isopropanol (Sigma-Aldrich). Absorption of reduced MTT and background signals was measured using a Spectramax M2 microplate reader at 570 nm and 670 nm, respectively. Control cells treated with buffer were used to calculate relative cell viability.

## Uptake of nanoparticles in vitro

To assess the uptake rate and intracellular localization of nanoparticles, cell lines were incubated with different concentrations of nanoparticles at 37°C or 4°C. Nanoparticles were loaded with 5 (6)-carboxyfluorescein (payload) and/or incorporated DiI in their phospholipid-membrane. Myrcludex B derived peptides were fluorescently labeled if necessary as indicated above. At the indicated time points, confocal laser scanning microscopy or flow cytometry were used for qualitative and quantitative analysis, respectively.

### Competitive inhibition experiments in vitro

NTCP-specific uptake of nanoparticles was investigated by pre-incubation with 400 nM free Myrcludex B fluorescently labeled with Atto-565 or Atto-488 as indicated.

### Binding mechanism studies in vitro

The hepatic cell dependent binding mechanism of nanoparticles was investigated by pre-incubation with 300 µg/mL heparin sulfate.

### Uptake mechanism studies on NTCP mediated internalization in vitro

The uptake mechanism of nanoparticles into HepG2 *SLC10A1* cells was investigated using different pharmacological pathway inhibitors as described previously (*Lunov et al., 2011*). Cells were pre-incubated using 100 µg/mL colchicine (micropinocytosis inhibitor), 10 µg/mL chlorpromazine (inhibitor of clathrin-mediated endocytosis), or 25 µg/mL nystatine (inhibitor of caveolin-mediated endocytosis) for 30 min before addition of nanoparticles.

### Confocal laser scanning microscopy

At indicated time points, cell nuclei were counterstained for 5 min using 1.0 µg/mL Hoechst 33342 (Sigma-Aldrich), washed with PBS and embedded using ProLong Gold antifading reagent (Invitrogen Life Technologies). For live cell imaging, cell nuclei were counterstained with Hoechst 33342, and if indicated cell membranes were stained with Cell Mask Deep Red Plasma Membrane Stain (1.0 µg/mL, Thermo Fisher Scientific) and NTCP was stained using fluorescently labeled Myrcludex B. Confocal laser scanning microscopy analysis was performed using an Olympus FV-1000 inverted microscope (Olympus Ltd.), equipped with a 60 × PlanApo N oil-immersion objective (numerical aperture 1.40).

## Flow cytometry analysis

To quantify the uptake rate of nanoparticles into non-hepatic and hepatic cell lines with different NTCP expression levels, flow cytometry analysis was performed. Cells were detached using 0.25% trypsin/EDTA (Sigma-Aldrich), washed twice with PBS and re-suspended in PBS containing 1% fetal calf serum, 0.05% NaN3, and 2.5 mM EDTA. At least 10,000 cells per setting were analyzed using a FACS Canto II flow cytometer (Becton Dickinson). Doublets were excluded and DiI or CF signals were measured. Relative mean fluorescence intensities (MFI) of DiI or CF signals normalized to untreated cells were calculated using Flow Jo VX software (TreeStar).

## High-content screening

To quantify the transfection of HepG2 cells deficient or expressing NTCP using DNA loaded nanoparticles, high-content screening was performed as described previously.(Lin et al., 2013) Cells were seeded in 96-well cell culture dishes, were allowed to adhere for 24 h, and treated with lipid nanoparticles at a DNA concentration of 1.5 µg/mL. To assess the transfection efficacy using high-content screening, cells were fixed with 3% paraformaldehyde 24 h post treatment and cell nuclei were counterstained with Hoechst 33342. Plates were scanned and analyzed using a Cellomics Array-Scan VTI (Thermo Scientific).

## Zebrafish embryo culture

Zebrafish embryos (*Danio rerio*) are a well-established vertebrate screening model for engineered nanomaterials (*Campbell et al., 2018*; *Einfalt et al., 2018*; *Sieber et al., 2017*). They were maintained in accordance with Swiss animal welfare regulations as described previously (*Sieber et al., 2017*). In brief, eggs from wild type ABC/TU and transgenic kdrl:EGFPs843 adult zebrafish were maintained in media at 28°C. Formation of pigment cells was prevented by 1-phenyl 2-thiourea (PTU, Sigma-Aldrich).

## Injection of nanoparticles into zebrafish embryos

To assess the systemic circulation of nanoparticles, samples were injected into transgenic kdrl:EGFPs843 zebrafish embryos (two dpf) as described previously (*Sieber et al., 2017*). In brief, calibrated volumes of 1 nL were injected into the duct of Cuvier of anesthetized and agarose-embedded zebrafish embryos using a micromanipulator (Wagner Instrumentenbau KG), a pneumatic Pico Pump PV830 (WPI), and a Leica S8APO microscope (Leica). The tail region of zebrafish embryos was imaged 1 h post injection (hpi) using an Olympus FV-1000 inverted confocal laser scanning microscope equipped with a 20 × UPlanSApo (numerical aperture 0.75) objective.

## Targeting of xenotransplanted human cells in the zebrafish model

Human HEK293 cells deficient or overexpressing *SLC10A1* were detached from 6-well cell culture dishes using 1 mL pre-warmed DMEM, washed (5 min at 200 g) and resuspended in 10 µL DMEM. Human cells (3 nL) were injected into the duct of Cuvier of ABC/TU zebrafish embryos. As soon as transgenic human cells stopped circulating and remained in the caudal vasculature tail region (after approximately two hpi), nanoparticles (1 nL) were injected as described above. Brightfield and fluorescence images of the tail region were taken 1 hpi of nanoparticles.

*Colocalization analysis.* Binding of nanoparticles to HEK293 cells was analyzed using the JaCoP plug-in Fiji. Therefore, Pearson´s Correlation Coefficient (PCC) was determined to assess the extent of colocalization (*Bolte and Cordelières, 2006*).

## Radioactive labeling of nanoparticles with $^{111}$In

Labeling of nanoparticles with $^{111}$In was performed with modifications as described previously (*van der Geest et al., 2015*). Nanoparticles were prepared as described above in PBS at a total lipid concentration of 60 mM (including 3 mol% DSPE-DTPA). Size exclusion chromatography was used to exchange the buffer system to citrate buffered saline pH 5.4, fractions were pooled and finally concentrated using Amicon Ultra-4 centrifugal filter units (100 kDa size exclusion). Nanoparticles (30 µmol) were incubated with 40 µl of $^{111}$InCl$_3$ (Mallinckrodt Pharmaceuticals) at 37°C for 45 min using a thermocycler. After incubation, $^{111}$In labeled nanoparticles were purified using NAP-5

columns (GE Healthcare) by elution with sterile saline (B. Braun Medical Inc). Fractions of 250 µL were collected and activity of each fraction was determined.

## Planar imaging of mice in vivo

All mice experiments were carried out in accordance with German legislation on animal welfare. Female NMRI mice (6–8 weeks) were obtained from Janvier Laboratories. For planar imaging, mice were anesthetized with Isoflurane (Baxter) and [111]In labeled nanoparticles with a total activity of 8–10 MBq (corresponding to 100 µL) were intravenously injected into the tail vein. Afterwards, the animals were placed in prone position (see *Figure 5—figure supplements 2A* and *3A*) on a planar gamma-imager (Biospace) equipped with a high energy collimator as described previously (*Müller et al., 2013*; *Wischnjow et al., 2016*). Images were recorded at the indicated time points with 10 min acquisition time.

## Planar imaging of harvested organs from mice and rats ex vivo

For planar imaging of organs, animals were anesthetized with Isoflurane (Baxter) and [111]In labeled nanoparticles were intravenously injected into the tail vein. Animals were sacrificed 15 min or 1 h post injection, organs were harvested and placed on a planar gamma-imager (Biospace) equipped with a high energy collimator. Images were recorded at the indicated time points with 10 min acquisition time.

## Quantitative organ biodistribution of nanoparticles in mice ex vivo

For biodistribution studies, [111]In labeled nanoparticles with a total activity of 1–2 MBq (corresponding to 100 µL) were intravenously injected into the tail vein of wild type mice. Animals were sacrificed (n = 3 per nanoparticle administration) 1 h post injection, organs were harvested and the radioactivity in each organ was measured with a Berthold LB 951G gamma counter. Each organ-associated activity was related to the injected dose. The percentage of injected dose (%ID) per organ was calculated using standard values for organ weights (*Mühlfeld et al., 2003*).

## Fluorescence imaging of nanoparticles in tissue cryo-sections

Nanoparticles incorporating 1 mol% DiI were intravenously injected into the tail vein of wildtype mice. Animals were sacrificed 1 h post injection and organs were snap-frozen in liquid nitrogen. Cryo-sections of 16 µm were mounted on Superfrost Plus Ultra microscope slides (Thermo Fisher Scientific) and counterstained with Hoechst 33342 (2 µg/mL). Slides were embedded in Prolong Gold Antifade Mountant (Thermo Fisher Scientific), sealed with nail polisher and analyzed using an Olympus FV-1000 inverted confocal laser scanning microscope equipped with a 40x UPlanFL N oil-immersion objective (numerical aperture 1.30).

## Immunohistochemistry of targeting ligand in tissue sections

After intravenous tail vein injection of nanoparticles, the mice were euthanized, organs were harvested, rinsed with PBS and immediately placed in a 4% formaldehyde solution in PBS. After fixation for 24 h, organs were dehydrated and embedded in paraffin. Sections of 5 µm thicknesses were cut using a microtome MICROM HM 355, placed onto a microscope slide and dried at 37˚C. After dewaxing and rehydration, epitope retrieval was performed. The primary antibody against Myr-preS2-31 (MA18/7, kind gift from Wolfram Gerlich) was added overnight at 4˚C, before incubation with the secondary antibody. Finally, slides were counterstained with hemalum (Merck KGaA) for 10 min, blued with tap water and mounted using Aquatex (Merck Millipore).

## Immunofluorescence imaging of liver cryo-sections

Animals were sacrificed 3 h post injection of nanoparticles and liver cryo-sections (16 µm) were prepared as described above. Slides were stained using primary antibodies against Myr-preS2-31 (MA18/7, 1:100 dilution) and *Slc10a1* (provided by Prof. Bruno Stieger, University Zürich, 1:100 dilution). Finally, cell nuclei were counterstained with Hoechst 33342 (2 µg/mL) and analyzed by confocal microscopy as described above.

## Statistical analysis

Statistical analysis for all experiments was performed by one-way analysis of variance (ANOVA) followed by Bonferroni post-hoc test using OriginPro 9.1 software (OriginLab Corporation). Differences between groups were considered to be statistically significant at the indicated p-values.

# Acknowledgements

We thank M Affolter, HG Belting and N Schellinx for providing zebrafish eggs, K Leotta for support with mice experiments, S Meßnard for sectioning of paraffin-embedded tissue, P Scheiffele and L Burklé for support with cryo-sectioning, and H Heerklotz and D Eckhardt for DSC and PPC measurements.

# Additional information

## Funding

| Funder | Grant reference number | Author |
| --- | --- | --- |
| Swiss National Science Foundation | 174975 | Dominik Witzigmann |
| Swiss National Science Foundation | 173057 | Jonas Buck Jörg Huwyler |
| Deutsche Forschungsgemeinschaft | 209091148 | Stephan Urban |
| German Center for Infection Research | 5.704 | Stephan Urban |
| German Center for Infection Research | 5.807 | Stephan Urban |
| Freiwillige Akademische Gesellschaft | FAG Basel | Dominik Witzigmann Sandro Sieber Jörg Huwyler |
| Stiftung zur Förderung des pharmazeutischen Nachwuchses in Basel | | Sandro Sieber |
| University of Basel | Novartis University Basel Excellence Scholarship for Life Sciences | Dominik Witzigmann |
| Novartis | Novartis University Basel Excellence Scholarship for Life Sciences | Dominik Witzigmann |

The funders had no role in study design, data collection and interpretation, or the decision to submit the work for publication.

## Author contributions

Dominik Witzigmann, Conceptualization, Resources, Formal analysis, Supervision, Funding acquisition, Validation, Visualization, Methodology, Writing—original draft, Project administration, Writing—review and editing; Philipp Uhl, Conceptualization, Formal analysis, Supervision, Funding acquisition, Validation, Investigation, Visualization, Methodology, Writing—original draft, Project administration, Writing—review and editing, Design and aquisition of rodent experiments; Sandro Sieber, Conceptualization, Formal analysis, Investigation, Methodology, Writing—review and editing, Design and aquisition of zebrafish experiments; Christina Kaufman, Conceptualization, Formal analysis, Investigation, Methodology, Writing—review and editing, Conception and design of HBV derived peptides and aquisition of rodent experiments; Tomaz Einfalt, Conceptualization, Formal analysis, Investigation, Methodology, Writing—original draft, Writing—review and editing, Physicochemical nanocarrier characterisation; Katrin Schöneweis, Formal analysis, Investigation, Methodology, Writing—original draft, Acquisition of tissue staining, Analysis and interpretation of

data; Philip Grossen, Formal analysis, Investigation, Methodology, Writing—original draft, Conception and design of nanocarrier; Jonas Buck, Investigation, Methodology, Writing—original draft, Acquisition of uptake mechanism studies; Yi Ni, Conceptualization, Resources, Methodology, Conception and design of NTC expressing cell lines; Susanne H Schenk, Conceptualization, Resources, Investigation, Methodology, Writing—original draft, Writing—review and editing, Design of DNA vectors; Janine Hussner, Resources, Formal analysis, Methodology, Contributed unpublished essential data; Henriette E Meyer zu Schwabedissen, Conceptualization, Methodology, Writing—review and editing, Design of NTCP encoding DNA vectors; Gabriela Québatte, Conceptualization, Investigation, Contributed unpublished essential data; Walter Mier, Stephan Urban, Conceptualization, Resources, Formal analysis, Funding acquisition, Writing—review and editing; Jörg Huwyler, Conceptualization, Resources, Formal analysis, Funding acquisition, Writing—original draft, Project administration, Writing—review and editing

### Author ORCIDs
Dominik Witzigmann https://orcid.org/0000-0002-8197-8558
Philip Grossen http://orcid.org/0000-0002-3416-5570
Henriette E Meyer zu Schwabedissen http://orcid.org/0000-0003-0458-4579
Jörg Huwyler https://orcid.org/0000-0003-1748-5676

### Ethics
Animal experimentation: Animal experimentation: Zebrafish embryo (*Danio rerio*) studies were performed in strict accordance with Swiss animal welfare regulations. Mouse and rat experiments were carried out in accordance with German legislation on animal welfare. All of the animals were handled according to approved institutional animal care and use protocol of the University of Basel and University of Heidelberg.

### Decision letter and Author response
Decision letter https://doi.org/10.7554/eLife.42276.041
Author response https://doi.org/10.7554/eLife.42276.042

## Additional files
### Supplementary files
• Transparent reporting form
DOI: https://doi.org/10.7554/eLife.42276.039

### Data availability
All data generated or analysed during this study are included in the manuscript and supporting files. Numerical data for all quantitative graphs are provided in the Figure source data.

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
