## [Decision Letter]

[Editors’ note: this article was originally rejected after discussions between the reviewers, but the authors were invited to resubmit after an appeal against the decision.]

Thank you for submitting your work entitled "Optimization-by-Design of a Hepatotropic Hepatitis B Virus-Mimetic Nanocarrier" for consideration by *eLife*. Your article has been reviewed by two peer reviewers, and the evaluation has been overseen by a Reviewing Editor and a Senior Editor. The following individuals involved in review of your submission have agreed to reveal their identity: Sjoerd Hak (Reviewer #2).

Our decision has been reached after consultation between the reviewers. Based on these discussions and the individual reviews below, we regret to inform you that your work will not be considered further for publication in *eLife*.

Your study reports a novel advance, using interesting methodologies. However, there are numerous concerns, specifically regarding experiments in animal models that preclude further consideration for publication. Also the applicability of the approach is somewhat overstated and not supported by the data presented.

*Reviewer #1:*

The manuscripts entitled 'Optimization-by-Design of a Hepatotropic Hepatitis B Virus-Mimetic Nanocarrier' describes investigations aimed at developing use of a myristolated preS peptide sequence derived from HBV to target liposomes to hepatocytes. Several experiments were carried out, which entailed analysis in cultured cells, zebrafish and mice. Although interesting, the work has significant shortcomings and is not suitable for publication in its present form. Some of the major concerns are the following:

1) Throughout the manuscript the nanoparticle is referred to as a 'virus mimetic'. However, the only similarity of the liposome to a virus is presence of the HBV-derived peptide. The description therefore seems to overstate the virus like properties. The particle does not have a capsid, nor does it carry any nucleic acid. Both are essential features of viruses.

2) Limitations of vectors used for delivery of nucleic acids are provided as justification for developing the liver targeting liposome. This point is re-emphasized in the conclusions where it is stated that the vector may have utility for treatment of monogenic diseases. However this opinion is not supported by the content of the manuscript. None of the experimentation characterizes properties of the vector when complexed to nucleic acids, nor is there any comparison to lipoplexes that use other targeting moieties, such as galactose.

3) In the experiments carried out on cultured cells, evidence that the payload is active or reaches the cytosol is not convincing. The punctate nature of the fluorescence suggests that the molecules may be trapped in the endosomes (Figure 1C).

4) Utility of the zebrafish model is minimal and the manuscript overstates the importance of the results from these experiments.

5) Data from analysis of the biodistribution of the particles and the radioactive payload in mice are particularly problematic.

a) Figure 6B indicates a thoracic location of the liver. Also images of mice in Figures 5 and 6 show dominant location of radioactivity in bilobed structures that look remarkably like lungs.

b) It is not clear whether the mice are prone or supine, which means that the left or right sides of the animals is unclear. Organ location is thus uncertain, and this adds to concerns about interpretation of the data

c) Evidence that the ^111^In was retained inside the nanoparticles is lacking. If the radioactivity diffused out of the particles images of the animals could reflect distribution of uncomplexed radioactivity.

Reviewer #2:

This study describes the development of liposomes conjugated with hepatitis B virus-derived peptides, which bind to the sodium-taurocholate co-transporting polypeptide (NTCP) on hepatocytes. Different forms of the peptide as well as densities of selected peptides were optimized in vitro with respect to NTCP binding, and in vivo in both zebrafish and mice with respect to pharmacokinetics and hepatocyte targeting.

I consider this a comprehensive study and the build-up of the manuscript is logical. The integrated use of the zebrafish model is appealing and it seems a useful tool in nanoparticle optimization studies. Overall, the study is well-done and is suitable for publication in *eLife*. I have one major issue, which relates to the data presented in Figure 5:

Animals were sacrificed 1 h post injection at which the blood pool levels varied significantly for the different formulations tested. Furthermore, the images reported in Figure 5D indicate very similar distributions in the liver of the 0.125 and 0.25 mol% formulations. Since differences in circulation time will affect targeting levels, the 0.125 mol% formulation may exhibit high liver targeting as well? Reporting ROI analysis of the in vivo images as well as ratios between for example blood/spleen/liver may be useful here.

[Editors’ note: what now follows is the decision letter after the authors submitted for further consideration.]

Thank you for sending your article entitled "Optimization-by-Design of a Hepatotropic Hepatitis B Virus-Mimetic Nanocarrier" for peer review at *eLife*. Your article is being evaluated by three peer reviewers, and the evaluation is being overseen by a Reviewing Editor and Didier Stainier as the Senior Editor.

Given the list of essential revisions, including new experiments, the editors and reviewers invite you to respond within the next two weeks with an action plan and timetable for the completion of the additional work. We plan to share your responses with the reviewers and then issue a binding recommendation.

1) Although the authors mentioned small molecule drugs, nucleic acids or proteins can be encapsulated and studied in appropriate disease models in the future. Lake of demonstration of further application may hamper the impact of this work. The authors should show the application of the carriers, which is a usual practice for high impact journals such as *eLife*. For example, does the liver-specific carrier increases plasmid DNA delivery and enhanced the transfection efficacy in livers sites?

2) The fact that ASGPR-targeted delivery systems can improve gene/drug delivery has been already well studied. Although the authors defined their drug carriers as alternative approaches. The advantage and limitation of the alternative approaches should be highlighted and discussed.

3) The anatomical data and their interpretation continue to be problematic. Shifting the outline of the animal(s) in the revised manuscript to align radioactivity with expected organ distribution is questionable and raises doubts about reliability of the data. Also, the location of the radioactivity does not correlate with murine anatomy and definition of organ structures. The spleen is shown as a lower abdominal organ, which is far removed from the liver on the transverse plane. This is not correct. Although information about imaging the mice in the prone position is now included, the images are presented unconventionally. Left and right sides, not indicated, are inverted. This issue has persisted. Please attempt to address it effectively. Perhaps consider conducting the imaging again and provide pictures if multiple animals – with multiple views.

4) Xenografted zebrafish are not widely employed in the field of vectorology, and arguments about the usefulness of the model are not compelling. You have to revise your text accordingly.

Reviewer #1:

Concerns about the manuscript remain in the revised version. The following are particular issues.

The anatomical data and their interpretation continue to be problematic. Shifting the outline of the animal(s) in the revised manuscript to align radioactivity with expected organ distribution is questionable and raises doubts about reliability of the data. Also, the location of the radioactivity does not correlate with murine anatomy and definition of organ structures. The spleen is shown as a lower abdominal organ, which is far removed from the liver on the transverse plane. This is not correct. Although information about imaging the mice in the prone position is now included, the images are presented unconventionally. Left and right sides, not indicated, are inverted.

Xenografted zebrafish are not widely employed in the field of vectorology, and arguments about the usefulness of the model are not compelling.

Reviewer #2:

All major concerns have been addressed satisfactory.

The demonstration of successful intracellular delivery of active nanoparticle payload (Figure 1—figure supplement 7 and 8) is not very convincing, the conclusion is based on images of 20 cells at most.

Although these images (Figure 1—figure supplement 7 and 8) indeed support the conclusion, more convincing results can be relatively easily obtained with very standard assays. However, taking the complete manuscript and focus of the study into account, I consider the manuscript complete, relevant, and suitable for publication in *eLife*.

Reviewer #3:

This is an interesting study about the use of HBV targeting properties to achieve liver-specific delivery. The authors developed a potent liver-specific drug delivery carrier modified with an alternative liver-targeting moiety. They utilized the zebrafish and murine model to optimize and characterize the ligand-modified liposome. The liver-specific delivery carrier showed increased liver uptake and decreased off-target delivery to other tissues in murine models. In general, the article describes a well performed study. There are some major concerns:

1) Although the authors mentioned small molecule drugs, nucleic acids or proteins can be encapsulated and studied in appropriate disease models in the future. Lake of demonstration of further application may hamper the impact of this work. The authors should show the application of the carriers, which is a usual practice for high impact journals such as *eLife*. For example, does the liver-specific carrier increases plasmid DNA delivery and enhanced the transfection efficacy in livers sites?

2) The fact that ASGPR-targeted delivery systems can improve gene/drug delivery has been already well studied. Although the authors defined their drug carriers as alternative approaches. The advantage and limitation of the alternative approaches should be highlighted and discussed.

---

## [Author Response]

[Editors’ note: the author responses to the first round of peer review follow.]

The rejection of our manuscript based on reviewer #2 is disappointing to us as the comments mainly criticized the term “virus-mimetic” and the mention of nucleic acid delivery as a future perspective. It is our opinion that both points addressed by the reviewer are not related to the quality or value of our experimental work but are in principle a matter of emphasis of the manuscript. The points addressed by reviewer #2 were now corrected in order to clarify the scope of our manuscript. In brief, we added new experiments, provide additional references, and rewrote the manuscript accordingly.

In addition, we feel that reviewer #2 did not take into account critical information already presented in our manuscript. We would like to point out that some of the concerns of the reviewer were already addressed based on presented data (Figure 1—figure supplement 6 and Figure 5B). These paragraphs were now rewritten to address the shortcomings highlighted by reviewer #2. Finally, we regret that recent headlines about research using the zebrafish as an early vertebrate in vivo model for nanoparticle screening became available during the review process only. We are convinced that this information would alleviate concerns of reviewer #2 related to the validity of our model and our present work was referenced accordingly. Based on the above points, we feel a strong need to contact you again. Considering the scope of your journal, our work would be of great interest for readers of *eLife*.

As discussed, we have directly addressed all reviewer comments, especially the concerns of reviewer #1.

Reviewer #1:The manuscripts entitled 'Optimization-by-Design of a Hepatotropic Hepatitis B Virus-Mimetic Nanocarrier' describes investigations aimed at developing use of a myristolated preS peptide sequence derived from HBV to target liposomes to hepatocytes. Several experiments were carried out, which entailed analysis in cultured cells, zebrafish and mice. Although interesting, the work has significant shortcomings and is not suitable for publication in its present form. Some of the major concerns are the following:1) Throughout the manuscript the nanoparticle is referred to as a 'virus mimetic'. However, the only similarity of the liposome to a virus is presence of the HBV-derived peptide. The description therefore seems to overstate the virus like properties. The particle does not have a capsid, nor does it carry any nucleic acid. Both are essential features of viruses.

We thank the reviewer for raising this concern. As the reviewer suggested, we modified the manuscript including title and figures. We replaced the term “virus-mimetic” throughout the manuscript with “nanoparticle” or variations such as “NTCP-targeted nanoparticles”. The “Title” was updated as follows:

Title: “Optimization-by-Design of Hepatotropic Lipid Nanoparticles Targeting the Sodium-Taurocholate Cotransporting Polypeptide”

2) Limitations of vectors used for delivery of nucleic acids are provided as justification for developing the liver targeting liposome. This point is re-emphasized in the conclusions where it is stated that the vector may have utility for treatment of monogenic diseases. However this opinion is not supported by the content of the manuscript. None of the experimentation characterizes properties of the vector when complexed to nucleic acids, nor is there any comparison to lipoplexes that use other targeting moieties, such as galactose.

We thank the reviewer for this comment. The major scope of this study was the development and in-depth optimization of NTCP-targeted nanoparticles using a unique approach of combining in vitroinvestigations and experiments in rodents with the emerging zebrafish model. Since the reviewer’s concern is not related to the validity of our study, we deleted statements regarding the delivery of nucleic acids and modified the “Introduction” and “Conclusion” section as follows:

Introduction: “In particular for the cell-type specific delivery of macromolecular therapeutic agents, selective targeting of parenchymal liver cells and internalization is needed.” AND “However, studies investigating alternative targeting strategies based on other hepatocyte-specific receptors are limited.”

Conclusion: “In particular, we see a great potential for our nanoparticle targeting strategy in the field ofmetabolic diseases of the liver.”

3) In the experiments carried out on cultured cells, evidence that the payload is active or reaches the cytosol is not convincing. The punctate nature of the fluorescence suggests that the molecules may be trapped in the endosomes (Figure 1C).

The comment of the reviewer, that there is no evidence that the payload is active or reaches the cytosol, ignores data presented in the Supplementary Information (Figure 1—figure supplement 8). We demonstrate the time-dependent internalization and toxicity of doxorubicin loaded nanoparticles into hNTCP overexpressing HepG2 cells. This result demonstrates that the payload is released into the cytosol resulting in cytotoxic effects. To avoid such misunderstanding, we emphasized this experiment as follows:

Results and Discussion: “Interestingly, Myr-preS2-31 modification enhanced the cytotoxic effects of propidium iodide and doxorubicin as compared to PEGylated nanoparticles (Figure 1—figure supplement 7 and 8) demonstrating that the payload is active and reaches the cytosol.”

In addition, we included an additional study performed using propidium iodide loaded nanoparticles. This data set confirms the release of the payload inside the cells resulting in nuclear accumulation of propidium iodide. An additional Figure (Figure 1—figure supplement 7) is presented. The manuscript was modified as follows:

Results and Discussion: “Of note, propidium iodide is a cell membrane impermeable drug. Thus, NTCP- targeted nanoparticles enabled internalization into cells and successful release into cytosol indicated by enhanced cytotoxic effects and nuclear counterstain.”

Legend Figure 1—figure supplement 7: “Time-dependent internalization and toxicity of propidium iodide loaded nanoparticles into hNTCP overexpressing HepG2 cells. Nanoparticles were passively loaded with propidium iodide (red signal). Representative confocal laser images for PEG nanoparticles and Myr-preS2-31 modified nanoparticles after specific time points are shown. Scale bar = 20 µm.”

Materials and methods: “Propidium iodide loading. For passive loading of propidium iodide into nanoparticles, a 10 mg/mL propidium iodide solution in PBS was used for the rehydration step of the homogenous lipid film. Resulting multilamellar vesicles were processed as described in the Materials and methods section including a final purification step.”

4) Utility of the zebrafish model is minimal and the manuscript overstates the importance of the results from these experiments.

We thank the reviewer for the critical comment indicating us that we did not highlight enough the breakthrough of this study in respect to state-of-the-art. In addition, we regret that recent headlines about our research using the zebrafish as an early vertebrate in vivomodel for nanoparticle screening became available during the review process only. The zebrafish has emerged as an early vertebrate in vivomodel for nanoparticle screening and our recent publications have been highlighted in several top-tier journals including ACS Nano and Journal of Controlled Release. In addition, we have an accepted original article in Nanomedicine:NBM and a review article in Advanced Drug Delivery Reviews. References were added accordingly. Recently, Shan et al. (doi: 10.1007/s13346-014-0210-2) reported huge discrepancies between in vitrosystems and rodent experiments during the development of targeted nanomedicines. Thus, there is a tremendous need to find innovative strategies for the development of cell-type specific delivery systems. It is important to mention, that the zebrafish model was not used as an alternative to rodent experiments but as a complementary in vivomodel to screen our nanocarriers. The zebrafish offers unique advantages: i) high reproducibility, ii) low costs, iii) high level of genetic homology to humans, iv) availability of transgenic lines, and v) most importantly optical transparency. This enables in vivoimaging at spatio-temporal resolution (i.e. down to a cellular level and at various time points). Consequently, we combined transgenic zebrafish lines with fluorescently labeled nanocarriers. Our approach offers the possibility to gain advanced insights into the circulation behavior and targeting properties of nanocarriers. We validated the data obtained in the zebrafish model in an established rodent model. To highlight the value of our work, that, we provide an additional reference and modified the “Introduction” and “Conclusion” sections as follows:

Introduction: “Recently, Shan et al. reported huge discrepancies between in vitro systems and rodent experiments during the development of targeted nanomedicines.(Shan et al., 2015) Therefore, we used the zebrafish as a complementary in vivo screening model based on our previous work. (Sieber et al., 2018; Campbell et al., 2018; Einfalt et al., 2018; Sieber et al., 2017) We assessed the effect of nanoparticles` ligand type and ligand density on their pharmacokinetics.” and “Strikingly, nanoparticles modified with 0.25 mol% Myr preS2 31 demonstrated a significant increase in the liver-to-kidney ratio confirming our conclusions from the zebrafish model, i.e. 0.25 mol% Myr-preS2-31 as an optimal ligand density (Figure 4B).”

Conclusions: “…and thus increase the accuracy of predictions for experiments in rodents”

5) Data from analysis of the biodistribution of the particles and the radioactive payload in mice are particularly problematic.a) Figure 6B indicates a thoracic location of the liver. Also images of mice in Figures 5 and 6 show dominant location of radioactivity in bilobed structures that look remarkably like lungs.b) It is not clear whether the mice are prone or supine, which means that the left or right sides of the animals is unclear. Organ location is thus uncertain, and this adds to concerns about interpretation of the datac) Evidence that the ^111^In was retained inside the nanoparticles is lacking. If the radioactivity diffused out of the particles images of the animals could reflect distribution of uncomplexed radioactivity.

We thank the reviewer for these comments regarding the biodistribution analysis.

a) This is a very important notion. We had the possibility to discuss this issue with experts in the field. Indeed, the presented qualitative data allows different interpretation. The dominant location of radioactivity in a bilobed structure in the abdominal cavity with an enhanced intensity in the right lobe suggesting the liver. Lungs would be expected to be symmetrical and localized in the thorax; see whole-body mouse atlas for comparison of organ localization (Baiker et al., 2008). In addition, quantitative biodistribution studies of harvested organs demonstrate that there is no accumulation in the lung. However, to clarify this point, we restrict our discussion to the quantitative biodistribution analysis. The planar imaging will just be used for informative purposes (now Figure 5—figure supplement 2). We adjusted Figure 5 and 6 accordingly and modified the manuscript as follows:

Results and Discussion: “One-hour post injection, plasma and organs were harvested to perform a quantitative biodistribution analysis ex vivo (Figure 5A). PEGylated nanoparticles showed the typical biodistribution of sterically stabilized nanoparticles with a strong signal in the blood (Figure 5A). […] Planar γ scintigraphy imaging of injected mice confirmed these observations (Figure 5—figure supplement 2).”

Legend Figure 5—figure supplement 2: “in vivo biodistribution and liver targeting of Myr-preS2-31 conjugated nanoparticles in mice. Nanoparticles were modified with different amounts of Myr-preS2-31. Static planar imaging of mice 15 min after intravenous injection of different ^111^In labeled nanoparticles with approximately 8 MBq.”

b) We included the position of the mice, provide an additional reference and modified the “Materials and methods” section as follows:

Materials and methods: “Afterwards, the animals were placed in prone position on a planar gamma-imager (Biospace) equipped with a high energy collimator as described previously.(Müller et al., 2013; Wischnjow et al., 2016)”

c) This comment of the reviewer ignores several aspects already presented in the manuscript, e.g. that ^111^In was never encapsulated inside the nanoparticles. Instead we used a lipid-chelator (i.e. DSPE-DTPA) to label the nanoparticle surface with ^111^In. In order to clarify this comment, we included a Discussion section highlighting the advantages of our labeling strategy. Two additional references are provided and the manuscript was modified as follows:

Results and Discussion: “Importantly, we incorporated DTPA-conjugated DSPE into the lipid bilayer to chelate ^111^In on the surface of nanoparticles. […] This offers an easy assessment to differentiate between non bound and nanoparticle bound ^111^In.

Reviewer #2:This study describes the development of liposomes conjugated with hepatitis B virus-derived peptides, which bind to the sodium-taurocholate co-transporting polypeptide (NTCP) on hepatocytes. Different forms of the peptide as well as densities of selected peptides were optimized in vitro with respect to NTCP binding, and in vivo in both zebrafish and mice with respect to pharmacokinetics and hepatocyte targeting.I consider this a comprehensive study and the build-up of the manuscript is logical. The integrated use of the zebrafish model is appealing and it seems a useful tool in nanoparticle optimization studies. Overall, the study is well-done and is suitable for publication in eLife. I have one major issue, which relates to the data presented in Figure 5:Animals were sacrificed 1 hour post injection at which the blood pool levels varied significantly for the different formulations tested. Furthermore, the images reported in Figure 5D indicate very similar distributions in the liver of the 0.125 and 0.25 mol% formulations. Since differences in circulation time will affect targeting levels, the 0.125 mol% formulation may exhibit high liver targeting as well? Reporting ROI analysis of the in vivo images as well as ratios between for example blood/spleen/liver may be useful here.

We thank the reviewer for this important suggestion. We calculated ratios between the blood pool levels and the three most important organs, i.e. liver (i.e. target organ), spleen (i.e. clearance organ), and kidney (i.e. control organ since nanoparticle bound ^111^In should not show renal excretion). An additional Figure (Figure 5—figure supplement 1) is presented and the manuscript was modified as follows:

Results and Discussion: “In order to highlight the ligand-density dependent hepatotropism, we calculated ratios between the blood pool and important organs, i.e. liver (i.e. target organ), spleen (i.e. clearance organ), and kidney (i.e. control organ since nanoparticle bound ^111^In should not show renal excretion) (Figure 5—figure supplement 1). Indeed, Myr preS2-31 modification ≥ 0.25 mol% resulted in increased liver/spleen-to-blood ratios.”

Legend Figure 5—figure supplement 1: “Organ ratios of ex vivo biodistribution analysis. Nanoparticles were modified with different amounts of Myr-preS2-31 and labeled with radioactive ^111^In. Quantitative biodistribution studies were performed 1 hour post injection. Radioactivity of each organ was determined with a γ-counter. Ratios of injected dose (%ID) per organ between the blood pool level and selected organs, i.e. liver (i.e. target organ), spleen (i.e. clearance organ), and kidney (i.e. control organ since nanoparticle bound ^111^In should not show renal excretion) were calculated. All values are shown as box plots of biological replicates (n = 3 independent experiments). *p < 0.05, **p < 0.01, ***p < 0.001.”

[Editors' note: the authors’ plan for revisions was approved and the authors made a formal revised submission.]

Reviewer #1:Concerns about the manuscript remain in the revised version. The following are particular issues.The anatomical data and their interpretation continue to be problematic. Shifting the outline of the animal(s) in the revised manuscript to align radioactivity with expected organ distribution is questionable and raises doubts about reliability of the data. Also, the location of the radioactivity does not correlate with murine anatomy and definition of organ structures. The spleen is shown as a lower abdominal organ, which is far removed from the liver on the transverse plane. This is not correct. Although information about imaging the mice in the prone position is now included, the images are presented unconventionally. Left and right sides, not indicated, are inverted.

The quantitative biodistribution studies of harvested organs demonstrated that nanoparticles modified with 0.25 mol% Myr-preS2-31 allowed highly efficient liver targeting while further increase in Myr-preS2-31 modification resulted in enhanced clearance by the spleen. In order to confirm this observation, we performed additional experiments in mice and rats. We injected ^111^In labeled nanoparticles and performed a planar γ scintigraphy analysis in vivo of the whole body and ex vivo of harvested organs. Photographical pictures were taken to indicate position and organ type. Rats were selected as additional species to increase the value of our study and to exclude species dependent effects. Three additional figures are provided (Figure 5—figure supplement 2-4), additional methods were added, and the manuscript was updated as follows:

Materials and methods: “Planar imaging of harvested organs from mice and rats ex vivo. For planar imaging of organs, animals were anesthetized with Isoflurane (Baxter) and ^111^In labeled nanoparticles were intravenously injected into the tail vein. Animals were sacrificed 15 min or 1 hour post injection, organs were harvested and placed on a planar gamma-imager (Biospace) equipped with a high energy collimator. Images were recorded at the indicated time points with 10 min acquisition time.”

Results and Discussion: “Planar gamma scintigraphy imaging of injected mice and harvested organs at various time points (15 min and 60 min) confirmed these observations (Figure 5—figure supplement 1, 2 and 3). […] Again, elevated Myr-preS2-31 modification has negative impacts on liver accumulation.”

Figure 5—figure supplement 1: “…Increase in Myr-preS2-31 modification (≥ 0.25 mol%) resulted in dominant location of radioactivity in a bilobed structure in the abdominal cavity with an enhanced intensity in the right lobe indicating the liver. In addition, nanoparticles with increased Myr-preS2-31 modification accumulated in an elongated structure in the far-left part of the abdomen under the liver, which is the spleen. For tissue identification and signal quantification see Figure 5A and Figure 5—figure supplement 2 and 3).”

Figure 5—figure supplement 2: “in vivo biodistribution and ex vivo organ distribution of PEGylated nanoparticles in mice. (A) Static planar imaging of mice in prone position 15 min after intravenous injection of different ^111^In labeled nanoparticles. Planar ex vivo imaging of harvested organs from mice (B) 15 min and (C) 60 min post injection.”

Figure 5—figure supplement 3: “in vivo biodistribution and ex vivo organ distribution of nanoparticles with elevated Myr-preS2-31 modification. (A) Static planar imaging of mice in prone position 15 min after intravenous injection of different ^111^In labeled nanoparticles. Planar ex vivo imaging of harvested organs from mice (B) 15 min and (C) 60 min post injection.”

Figure 5—figure supplement 4: “Organ biodistribution of different nanoparticles in rats. Planar ex vivo imaging of harvested organs from rats injected with (A) PEGylated nanoparticles or (B) nanoparticles with elevated Myr-preS2-31 modification was performed 60 min post injection.”

Xenografted zebrafish are not widely employed in the field of vectorology, and arguments about the usefulness of the model are not compelling.

We agree with the reviewer that zebrafish are not yet widely employed in the field of vectorology. In the present work, we therefore combine results in the zebrafish with data from two additional species (mouse and rat). Furthermore, we would like to point out that zebrafish is an emerging screening model for nanomedicines. This is increasingly recognized by other experts in the field. Our recent published review article “Zebrafish as a Preclinical in vivo Screening Model for Nanomedicines” was selected by the Editor for the Editors’ Collection issue by Advanced Drug Delivery Reviews. It is one of the highlights of our study that the observations made in the zebrafish are highly predictive for experiments in rodents. To address the concern of the reviewer regarding the use of xenografted zebrafish, we added eight additional references and updated the “Discussion” section to highlight advantages and disadvantages of xenotransplanted zebrafish as follows:

Results and Discussion: “In recent years, several groups have used xenografted zebrafish for various investigations including the assessment of nanoparticles.(Sieber et al., 2019; Evensen et al., 2016; Wertman et al., 2016; Brown et al., 2017; He et al., 2012; Lin et al., 2017; Veinotte et al., 2014; Wagner et al., 2010) Despite anatomical differences with mammals, zebrafish xenotransplantation models are an emerging preclinical tool offering several practical advantages as compared to mouse xenografting models including prolific reproduction, facilitated xenotransplantation (no immune rejection due to limited adaptive immune response), and optical transparency enabling high throughput screening. For our study,…”

Reviewer #3:This is an interesting study about the use of HBV targeting properties to achieve liver-specific delivery. The authors developed a potent liver-specific drug delivery carrier modified with an alternative liver-targeting moiety. They utilized the zebrafish and murine model to optimize and characterize the ligand-modified liposome. The liver-specific delivery carrier showed increased liver uptake and decreased off-target delivery to other tissues in murine models. In general, the article describes a well performed study. There are some major concerns:1) Although the authors mentioned small molecule drugs, nucleic acids or proteins can be encapsulated and studied in appropriate disease models in the future. Lake of demonstration of further application may hamper the impact of this work. The authors should show the application of the carriers, which is a usual practice for high impact journals such as eLife. For example, does the liver-specific carrier increases plasmid DNA delivery and enhanced the transfection efficacy in livers sites?

The scope of this study is the development and in-depth optimization of NTCP-targeted nanoparticles using a unique approach of combining in vitro investigations and experiments in rodents with the emerging zebrafish model. in vivo liver transfection studies are beyond the scope of this manuscript. However, we performed an in vitro study to investigate the potential application of NTCP-targeted lipid nanoparticles as gene delivery systems. We successfully entrapped a DNA vector coding for GFP into lipid nanoparticles and modified their surface with Myr-preS2-31. These systems were tested in HepG2 SLC10A1 cells regarding their gene delivery efficacy. An additional figure was provided (Figure 1—figure supplement 9), additional methods and references were added, and the manuscript was updated as follows:

Results and Discussion: “In order to demonstrate the potential application of Myr-preS2-31 modified nanoparticles as drug delivery system, we successfully incorporated small molecular payloads as well as larger compounds into nanoparticles payloads (i.e. propidium iodide, doxorubicin, FITC-labeled peptide, DNA vector) to enhance their internalization into NTCP expressing cells (Figure 1—figure supplement 6, 7, 8 and 9) […] To investigate the potential application of NTCP-targeted lipid nanoparticles as gene delivery systems, we encapsulated a DNA vector coding GFP into lipid nanoparticles based on a clinically approved lipid composition and modified their surface with Myr-preS2-31. High content screening analysis demonstrated that modification of nanoparticles with Myr-preS2-31 significantly increases the transfection of NTCP expressing cells (Figure 1—figure supplement 9).”

Materials and methods: “DNA vector loading. Lipid nanoparticles entrapping DNA were prepared as previously described with modifications.(Kulkarni et al., 2019, 2018, 2017) Briefly, lipids (ionizable lipid, cholesterol, DSPC, DSPE-PEG2000, and DSPE-PEG2000-Maleimide at a molar ratio of 50:39:10:0.75:0.25 mol%) were dissolved in ethanol at a total lipid concentration of 15 mM. The DNA vector was dissolved in 25 mM sodium acetate (pH 4) at an N/P ratio of 6). After T-junction mixing at a flow rate ratio of 3:1 v/v, the pH was neutralized using a 5x excess of D-PBS, the appropriate amount of Myr-preS2-31 was added, and the resulting suspension was dialyzed against D-PBS to remove residual ethanol.”

“High-content screening. To quantify the transfection of HepG2 cells deficient or expressing NTCP using DNA loaded nanoparticles, high-content screening was performed as described previously.(Lin et al., 2013) Cells were seeded in 96-well cell culture dishes, were allowed to adhere for 24 h, and treated with lipid nanoparticles at a DNA concentration of 1.5 μg/mL. To assess the transfection efficacy using high-content screening, cells were fixed with 3% paraformaldehyde 24 h post treatment and cell nuclei were counterstained with Hoechst 33342. Plates were scanned and analyzed using a Cellomics ArrayScan VTI (Thermo Scientific).”

Figure 1—figure supplement 9: “Activity of DNA loaded lipid nanoparticles (LNP). LNP entrapping GFP coding DNA were modified with Myr-preS2-31 and compared to non-modified PEG DNA-LNP. (A) Representative fluorescence images of HepG2 SLC10A1 cells 24 h after PEG DNA-LNP and Myr-preS2-31 DNA-LNP treatment are shown. Blue signal: Hoechst stain of cell nuclei. Green signal: GFP expressing cells. CellOmics analysis was performed to quantify transfection efficiency. (B) Quantification of transfection efficiency. All values are shown as mean ± SD of biological replicates (n = 4 experiments). **p < 0.01.”

2) The fact that ASGPR-targeted delivery systems can improve gene/drug delivery has been already well studied. Although the authors defined their drug carriers as alternative approaches. The advantage and limitation of the alternative approaches should be highlighted and discussed.

We thank the reviewer for this comment. To clarify this point, we modified the conclusion section and included important drawbacks of ASGPR and LDLR-targeted delivery systems and illustrated a possible advantage of our system as follows:

Conclusion: “Despite the fact that current liver targeting strategies such as ASGPR- or LDLR-targeted delivery systems have already demonstrated improved drug and gene delivery in various preclinical models, these systems also suffer from certain drawbacks including complex synthesis of multivalent glycans or strong evidence that a majority of endocytosed gene carriers is recycled back to the cell exterior thereby reducing activity.(Witzigmann et al., 2016; Sahay et al., 2013) Due to the availability of efficient of peptide manufacturing protocols, the proposed NTCP targeted delivery platform is an alternative and promising approach which warrants further investigation. For future clinical applications, optimized Myr-preS2-31 conjugated nanoparticles entrapping small molecule drugs, nucleic acids or proteins need to be studied in appropriate animal models of disease.”